

# q-Rung orthopair fuzzy dynamic aggregation operators with time sequence preference for dynamic decision-making

Hafiz Muhammad Athar Farid[1], Muhammad Riaz[1], Vladimir Simic[2,3] and Xindong Peng[4]

[1] University of the Punjab, Lahore, Pakistan
[2] Faculty of Transport and Traffic Engineering, University of Belgrade, Belgrade, Serbia
[3] Department of Industrial Engineering and Management, Yuan Ze University, Taoyuan City, Taiwan
[4] School of Information Engineering, Shaoguan University, China, Shaoguan, China

Corresponding author
Hafiz Muhammad Athar Farid,
hmatharfarid@gmail.com

## ABSTRACT

The q-rung orthopair fuzzy set (q-ROPFS) is a kind of fuzzy framework that is capable of introducing significantly more fuzzy information than other fuzzy frameworks. The concept of combining information and aggregating it plays a significant part in the multi-criteria decision-making method. However, this new branch has recently attracted scholars from several domains. The goal of this study is to introduce some dynamic q-rung orthopair fuzzy aggregation operators (AOs) for solving multi-period decision-making issues in which all decision information is given by decision makers in the form of "q-rung orthopair fuzzy numbers" (q-ROPFNs) spanning diverse time periods. Einstein AOs are used to provide seamless information fusion, taking this advantage we proposed two new AOs namely, "dynamic q-rung orthopair fuzzy Einstein weighted averaging (DQROPFEWA) operator and dynamic q-rung orthopair fuzzy Einstein weighted geometric (DQROPFEWG) operator". Several attractive features of these AOs are addressed in depth. Additionally, we develop a method for addressing multi-period decision-making problems by using ideal solutions. To demonstrate the suggested approach's use, a numerical example is provided for calculating the impact of "coronavirus disease" 2019 (COVID-19) on everyday living. Finally, a comparison of the proposed and existing studies is performed to establish the efficacy of the proposed method. The given AOs and decision-making technique have broad use in real-world multi-stage decision analysis and dynamic decision analysis.

## INTRODUCTION

Multi-criteria decision making (MCDM) is broadly used in the scientific disciplines of societal structure, economic growth, strategic planning, engineering and among others. It is a decision-making process that involves selecting a preferable solution from a finite set of conceivable alternatives that have been evaluated on numerous qualitative or quantitative features by multiple DMs. Due to the intricacy and unpredictability of the problem, the time constraints, and the limited competence of the participants in the MCDM, DMs

occasionally do not deliver their assessment conclusions in the form of accurate values. Uncertainty is among the issues that have arisen as a result of dealing with genuine circumstances in engineering and scientific knowledge. Zadeh's "fuzzy set" (FS) theory is an effective tool for depicting the world's unpredictability and ambiguity (*Zadeh, 1965*). To have a greater understanding of the objective world's uncertainty and therefore to be capable of explaining it, several extensions to this theory have been proposed. In 1986, Atanassov modified Zadeh's fuzzy set theory and introduced the "intuitionistic fuzzy set" (IFS) theory (*Atanassov, 1986*). According to FSs, Atanassov's IFS theory provides a more powerful strategy for dealing with ambiguity and uncertainties. IFS offers two types of degrees: "membership degree (MSD) and non-membership degree" (NMSD). As a result, one may argue that IFS is better suited for representing DM's perspectives in decision-making. As a result, IFS has been used to a variety of MCDM issues, including supply chain, medical diagnostics and decision-making. When the total of the MSD and NMSD is not in the range $[0, 1]$, as in $0 < 0.53 + 0.72 = 1.25 \nleq 1$, this sort of problem cannot be handled using IFS. To address this type of challenge, *Yager (2014)* developed the "Pythagorean fuzzy set" (PyFS) as an extension of IFS, in which the sum of the squares of the MSD and NMSD equals "<1" or "=1". Since then, PyFS has gained increasing attention as a result of its characteristics. *Rani, Mishra & Mardani (2020)* investigated the evaluation of pharmaceutical therapies for type 2 diabetes mellitus in PyFS data using the new entropy and scoring functions. *Garg, (2017)* suggested an extended PyFS information accumulation technique based on Einstein norms and used it to MCDM applications. *Jana, Senapati & Pal (2019)* employed PyFS information-based solution principles and Pythagorean Dombi AOs to solve MCDM challenges. *Liang et al. (2019)* designed a decision-making system for evaluating product quality in the online banking sector based on Pythagorean fuzzy operational scientific principles. *Liang et al. (2018)* implemented the expanded "Bonferroni mean" AOs in PyFS and then constructed an algorithm to implement the proposed strategy. While IFS and PyFS are capable of resolving some unclear circumstances, they cannot handle all sorts of data completely. As seen in this example, when a DM employed 0.81 as the MSD and 0.72 as the NMSD, $0.81^2 + 0.72^2 = 1.1745 \nleq 1$. As a result, PyFS is incapable of dealing with such uncertainty. *Yager (2017)* proposed q-ROPFSs to solve these challenges, which are more resilient and common than IFS and PyFS. q-ROPFSs can be used to solve complex and uncertain problems in fuzzy frames. Additionally, *Liu & Wang (2018)* presented q-ROPF aggregating functions and demonstrated their use in solving the MCDM issue. *Tang, Chiclana & Liu (2020)* introduced the rough set approach for q-ROPFSs with applications to stock investment evaluation.

AOs are useful mechanisms for combining all input arguments into a single fully integrated value, notably in the MCDM analysis. *Krishankumar et al. (2020)* proposed generalized "Maclaurin symmetric mean" AOs and *Liu, Chen & Wang (2020)* gave the notion of "power Maclaurin symmetric mean" AOs for q-ROPFNs. *Kumar & Gupta (2023)* introduced some q-ROPF normal basic AOs merging with confidence level concept. *Liu et al. (2022)*, *Kumar & Chen (2022)*, *Attaullah et al. (2022)*, *Garg et al. (2022)*, *Riaz et al. (2021)*, *Farid & Riaz (2021)* and *Wei, Gao & Wei (2018)* proposed some extensive AOs for q-ROPFSs and their hybrid structure.

The prior work, in general, centred on the development of models for gathering q-ROPF information over the same time span. However, in many difficult cases requiring decision-making, it is necessary to take into account how various options perform over the course of time. Due to the fact that these dilemmas include the assortment of data at distinct time frames within a period, they are classified as multi-period decision-making (MPDM) issues. In the last few decades, a large number of researchers have investigated the temporal generalised variations (also frequently referred to as dynamic) of existing fuzzy AOs and studied the efficiency with which they function in the MPDM. *Yang et al. (2017)*, *Kamaci, Petchimuthu & Akcetin (2021)*, *Peng & Wang (2014)*, *Gumus & Bali (2017)* and *Jana, Pal & Liu (2022)* gave some dynamic AOs for the different extension of FS. Some extensive work related to AOs can be seen in *Dabic-Miletic & Simic (2023)*, *Naseem et al. (2023)*, *Abid & Saqlain (2023)*. *Jana & Pal (2021)* proposed dynamical hybrid method to design decision making process. Some AOs related to q-ROPF soft information can be seen in *Hayat et al. (2023)*, *Yang et al. (2022)*. Linear Diophantine fuzzy soft-max AOs and numerically validated approach to modeling water hammer phenomena is given in *Riaz & Farid (2023)*, *Kausar, Farid & Riaz (2023)*. More work related to proposed idea can be seen in *Liu et al. (2023)*, *Liu, Li & Lin (2023)*, *Zhang et al. (2022b)*.

There are a number of different pairings of t-norms and t-conorms that may be found in order to produce q-ROPFS intersections and unions. Einstein's t-norms and t-conorms are appropriate options for determining the product and sum of q-ROPFSs, respectively. Fluent algebraic product and sum techniques may be obtained through the use of Einstein product and sum, which are respectively characterised in terms of Einstein t-norms and t-conorms. In addition, numerous different MCDM strategies integrate alternative evaluations within the allotted window of time. In point of fact, the process of evaluation need to take into consideration not only the performance of alternatives in the here and now, but also the performance of alternatives in the past. The ideal choice is determined by considering both the alternatives' historical and their current performance in relation to specific MCDM problems (*Dong et al., 2024*).

As a consequence of this, the major purpose of this article is to construct some dynamic AOs based on Einstein operations on q-ROPFSs. We present the dynamic q-ROPF Einstein averaging and geometric operators for this aim in this research. Einstein operations in q-ROPFSs are used to collect data across a wide range of time periods and aggregate it into a single value. This is what sets them apart from other approaches. We investigate important facets of these operators, such as their idempotence, boundedness, and monotonicity, among other things.

The remaining parts of the article are structured as described below. In 'Fundamental concepts', we will cover the fundamentals of q-ROPFS, in addition to several other significant concepts. In 'Dynamic Q-rung orthopair fuzzy einstein AOS', you'll find various dynamic q-ROPF Einstein AOs, each with their own set of alluring characteristics. In the section 'MCDM methods with proposed AOS', we build an MCDM technique using the AOs that were described. In 'Case study', a comprehensive discussion of the case study is presented, complete with numerical figures and a contrast to the AOs now in effect. The most important findings of the study are discussed in 'Conclusion'.

# FUNDAMENTAL CONCEPTS

This part provides an overview of the fundamental principles pertaining to q-ROPFSs.

**Definition 2.1** *(Yager, 2017) A q-ROPFS $W$ on $X$ is given as*

$W = \{\langle \aleph, \mu^{\zeta}{}_W(\aleph), \nu^{\zeta}{}_W(\aleph) \rangle : \aleph \in X\}$

*here $\mu^{\zeta}{}_W, \nu^{\zeta}{}_W : X \to [0,1]$ denote the MSD and NMSD of the alternative $\aleph \in X$ and $\forall \aleph$ we have*

$0 \leq \mu^{\zeta q}{}_W(\aleph) + \nu^{\zeta q}{}_W(\aleph) \leq 1.$

*Moreover, $\pi_W(\aleph) = \sqrt[q]{1 - \mu^{\zeta q}{}_W(\aleph) - \nu^{\zeta q}{}_W(\aleph)}$ is called the "indeterminacy degree" of $x$ to $W$.*

*Liu & Wang (2018)* proposed that several operations on q-ROPFNs be performed using the provided concepts.

**Definition 2.2** *(Liu & Wang, 2018) Consider $\alpha_1^{\aleph} = \langle \mu^{\zeta}{}_1, \nu^{\zeta}{}_1 \rangle$ and $\alpha_2^{\aleph} = \langle \mu^{\zeta}{}_2, \nu^{\zeta}{}_2 \rangle$ are the two q-ROPFNs and $\sigma > 0$, then*

(1) $(\alpha_1^{\aleph})^c = \langle \nu^{\zeta}{}_1, \mu^{\zeta}{}_1 \rangle;$

(2) $\alpha_1^{\aleph} \wedge \alpha_2^{\aleph} = \langle \min\{\mu^{\zeta}{}_1, \nu^{\zeta}{}_1\}, \max\{\mu^{\zeta}{}_2, \nu^{\zeta}{}_2\} \rangle;$

(3) $\alpha_1^{\aleph} \vee \alpha_2^{\aleph} = \langle \max\{\mu^{\zeta}{}_1, \nu^{\zeta}{}_1\}, \min\{\mu^{\zeta}{}_2, \nu^{\zeta}{}_2\} \rangle;$

(4) $\alpha_1^{\aleph} \oplus \alpha_2^{\aleph} = \langle \sqrt[q]{\mu^{\zeta q}{}_1 + \mu^{\zeta q}{}_2 - \mu^{\zeta q}{}_1 \mu^{\zeta q}{}_2}, \nu^{\zeta}{}_1 \nu^{\zeta}{}_2 \rangle;$

(5) $\alpha_1^{\aleph} \otimes \alpha_2^{\aleph} = \langle \mu^{\zeta}{}_1 \mu^{\zeta}{}_2, \sqrt[q]{\nu^{\zeta q}{}_1 + \nu^{\zeta q}{}_2 - \nu^{\zeta q}{}_1 \nu^{\zeta q}{}_2} \rangle;$

(6) $\sigma \alpha_1^{\aleph} = \langle \sqrt[q]{1 - (1 - \mu^{\zeta q}{}_1)^{\sigma}}, \nu^{\zeta \sigma}{}_1 \rangle;$

(7) $(\alpha_1^{\aleph})^{\sigma} = \langle \mu^{\zeta \sigma}{}_1, \sqrt[q]{1 - (1 - \nu^{\zeta q}{}_1)^{\sigma}} \rangle.$

**Definition 2.3** *(Liu & Wang, 2018) Assume that $\alpha^{\aleph} = \langle \mu^{\zeta}, \nu^{\zeta} \rangle$ is the q-ROPFN, then its "score function" (SF) $S^{\top}$ of $\alpha^{\aleph}$ is defined as*

$S^{\top}(\alpha^{\aleph}) = \mu^{\zeta q} - \nu^{\zeta q}, \quad S^{\top}(\alpha^{\aleph}) \in [-1, 1].$

**Definition 2.4** *(Liu & Wang, 2018) Assume that $\alpha^{\aleph} = \langle \mu^{\zeta}, \nu^{\zeta} \rangle$ is the q-ROPFN, then its "accuracy function" (AF) $H^{\top}$ of $\alpha^{\aleph}$ is characterized as*

$H^{\top}(\alpha^{\aleph}) = \mu^{\zeta q} + \nu^{\zeta q}, \quad H^{\top}(\alpha^{\aleph}) \in [0, 1].$

*Riaz et al. (2020)* presented the Einstein operations for q-ROPFNs and explored the desired characteristics of these operations. They developed multiple AOs with the assistance of these operations.

**Definition 2.5** *(Riaz et al., 2020) Let $\alpha_1^{\aleph} = \langle \mu^{\zeta}{}_1, \nu^{\zeta}{}_1 \rangle$ and $\alpha_2^{\aleph} = \langle \mu^{\zeta}{}_2, \nu^{\zeta}{}_2 \rangle$ be q-ROPFNs, $\zeta > 0$ be real number, then*

(1) $\overline{\alpha_1^{\aleph}} = \langle \nu^{\zeta}{}_1, \mu^{\zeta}{}_1 \rangle$

(2) $\alpha_1^{\aleph} \vee_{\epsilon} \alpha_2^{\aleph} = \langle \max\{\mu^{\zeta}{}_1, \mu^{\zeta}{}_2\}, \min\{\nu^{\zeta}{}_1, \nu^{\zeta}{}_2\} \rangle$

(3) $\alpha_1^{\aleph} \wedge_{\epsilon} \alpha_2^{\aleph} = \langle \min\{\mu^{\zeta}{}_1, \mu^{\zeta}{}_2\}, \max\{\nu^{\zeta}{}_1, \nu^{\zeta}{}_2\} \rangle$

(4) $\alpha_1^{\aleph} \otimes_{\epsilon} \alpha_2^{\aleph} = \left\langle \dfrac{\mu^{\zeta}{}_{1 \cdot \epsilon} \mu^{\zeta}{}_2}{\sqrt[q]{1 + (1 - \mu^{\zeta q}{}_1)_{\cdot \epsilon} (1 - \mu^{\zeta q}{}_2)}}, \sqrt[q]{\dfrac{\nu^{\zeta q}{}_1 + \nu^{\zeta q}{}_2}{1 + \nu^{\zeta q}{}_{1 \cdot \epsilon} \nu^{\zeta q}{}_2}} \right\rangle$

(5) $\alpha_1^{\aleph} \oplus_{\epsilon} \alpha_2^{\aleph} = \left\langle \sqrt[q]{\dfrac{\mu^{\zeta q}{}_1 + \nu^{\zeta q}{}_2}{1 + \mu^{\zeta q}{}_{1 \cdot \epsilon} \mu^{\zeta q}{}_2}}, \dfrac{\nu^{\zeta}{}_{1 \cdot \epsilon} \nu^{\zeta}{}_2}{\sqrt[q]{1 + (1 - \nu^{\zeta q}{}_1)_{\cdot \epsilon} (1 - \nu^{\zeta q}{}_2)}} \right\rangle$

**Table 1  Comparative analysis of q-ROPFNs.**

| Theories | Merits | Limitations |
|---|---|---|
| Fuzzy sets (*Zadeh, 1965*) | Allocate MSD in $[0, 1]$ | Can not allocate NMSD |
| IFSs (*Atanassov, 1986*) | Allocate both MSD and NMSD, | Fails when MSD+NMSD $> 1$ |
| PyFSs (*Yager, 2014*) | Allocate both MSD and NMSD, | Fails when MSD$^2$ + NMSD$^2 > 1$ |
| | superior than the IFNs | |
| q-ROPFSs (*Yager, 2017*) | Allocate both MSD and NMSD, | Can not deal with MSD$^q$ + NMSD$^q > 1$ |
| | superior than IFNs, PFNs, | and MSD = NMSD = 1 |
| | a broader space for MSD and NMSD & | |

$$(6) \quad \zeta \cdot_\epsilon \alpha_1^\aleph = \left\langle \sqrt[q]{\frac{(1+(\mu^\zeta{}_1)^q)^\zeta - (1-(\mu^\zeta{}_1)^q)^\zeta}{(1+(\mu^\zeta{}_1)^q)^\zeta + (1-(\mu^\zeta{}_1)^q)^\zeta}}, \frac{\sqrt[q]{2}(\nu^\zeta{}_1)^\zeta}{\sqrt[q]{(2-(\mu^\zeta{}_1)^q)^\zeta + ((\nu^\zeta{}_1)^q)^\zeta}} \right\rangle$$

$$(7) \quad \alpha_1^\aleph \zeta = \left\langle \frac{\sqrt[q]{2}(\mu^\zeta{}_1)^\zeta}{\sqrt[q]{(2-(\mu^\zeta{}_1)^q)^\zeta + ((\mu^\zeta{}_1)^q)^\zeta}}, \sqrt[q]{\frac{(1+(\nu^\zeta{}_1)^q)^\zeta - (1-(\nu^\zeta{}_1)^q)^\zeta}{(1+(\nu^\zeta{}_1)^q)^\zeta + (1-(\nu^\zeta{}_1)^q)^\zeta}} \right\rangle$$

**Theorem 2.6**  (*Riaz et al., 2020*) *Let* $\alpha_1^\aleph$ *and* $\alpha_2^\aleph$ *be q-ROPFNs and* $\zeta, \zeta_1, \zeta_2 \geq 0$ *be any real number, then*

$$(1) \quad \alpha_2^\aleph \otimes_\epsilon \alpha_1^\aleph = \alpha_1^\aleph \otimes_\epsilon \alpha_2^\aleph$$
$$(2) \quad \alpha_2^\aleph \oplus_\epsilon \alpha_1^\aleph = \alpha_1^\aleph \oplus_\epsilon \alpha_2^\aleph$$
$$(3) \quad (\alpha_2^\aleph \otimes_\epsilon \alpha_1^\aleph)^\zeta = \alpha_2^\aleph \zeta \otimes_\epsilon \alpha_1^\aleph \zeta$$
$$(4) \quad \zeta \cdot_\epsilon (\alpha_1^\aleph \oplus_\epsilon \alpha_2^\aleph) = \zeta \cdot_\epsilon \alpha_1^\aleph \oplus_\epsilon \zeta \cdot_\epsilon \alpha_2^\aleph$$
$$(5) \quad \alpha_1^\aleph \zeta_1 \otimes_\epsilon \alpha_1^\aleph \zeta_2 = \alpha_1^\aleph \zeta_1 + \zeta_2$$
$$(6) \quad \zeta_1 \cdot_\epsilon (\zeta_2 \cdot_\epsilon \alpha_1^\aleph) = (\zeta_1 \cdot_\epsilon \zeta_2) \cdot_\epsilon \alpha_1^\aleph$$
$$(7) \quad (\alpha_1^\aleph \zeta_1) \zeta_2 = (\alpha_1^\aleph) \zeta_1 \cdot_\epsilon \zeta_2$$
$$(8) \quad \zeta_1 \cdot_\epsilon \alpha_1^\aleph \oplus_\epsilon \zeta_2 = (\zeta_1 + \zeta_2) \cdot_\epsilon \alpha_1^\aleph$$

## Superiority of q-ROPFNs and comparison with other fuzzy numbers

An effective solution for problems requiring machine learning, fuzzy computing, and MCDM may be found in the extended MSD and NMSD of q-ROPFNs. The performance of a q-ROPFN is superior than that of other fuzzy numbers (FNs), IFNs and PFNs. The advantages and disadvantages of q-ROPFNs in contrast to those of other fuzzy numbers are outlined in detail in the Table 1. The geometrical depiction of q-ROPFS with IFS and PFS is shown in Fig. 1.

## DYNAMIC Q-RUNG ORTHOPAIR FUZZY EINSTEIN AOS

Following is a discussion of certain dynamic q-ROPF Einstein AOs and their attractive characteristics.

### DQROPFEWA operator

**Definition 3.1**  *Consider* $\alpha^\aleph(\beth_k) = \left( \mu^\zeta{}_{\alpha^\aleph(m_k)}, \nu^\zeta{}_{\alpha^\aleph(m_k)} \right)$ $(k = 1, \ldots, d)$ *the assortment of q-ROPF values for $d$ distinct time periods* $(k = 1, 2, \ldots, d)$. $\gamma^\beta(k) = \left[ \gamma^\beta(m_1), \gamma^\beta(m_2), \ldots, \gamma^\beta(m_d) \right]^T$ *is the weight vector (WV) of the periods, where* $\sum_{k=1}^d \gamma^\beta(m_k) = 1$ *and let* $DQROPFEWA : X^n \to X$. *If* $DQROPFEWA(\alpha^\aleph(m_1), \alpha^\aleph(m_2),$

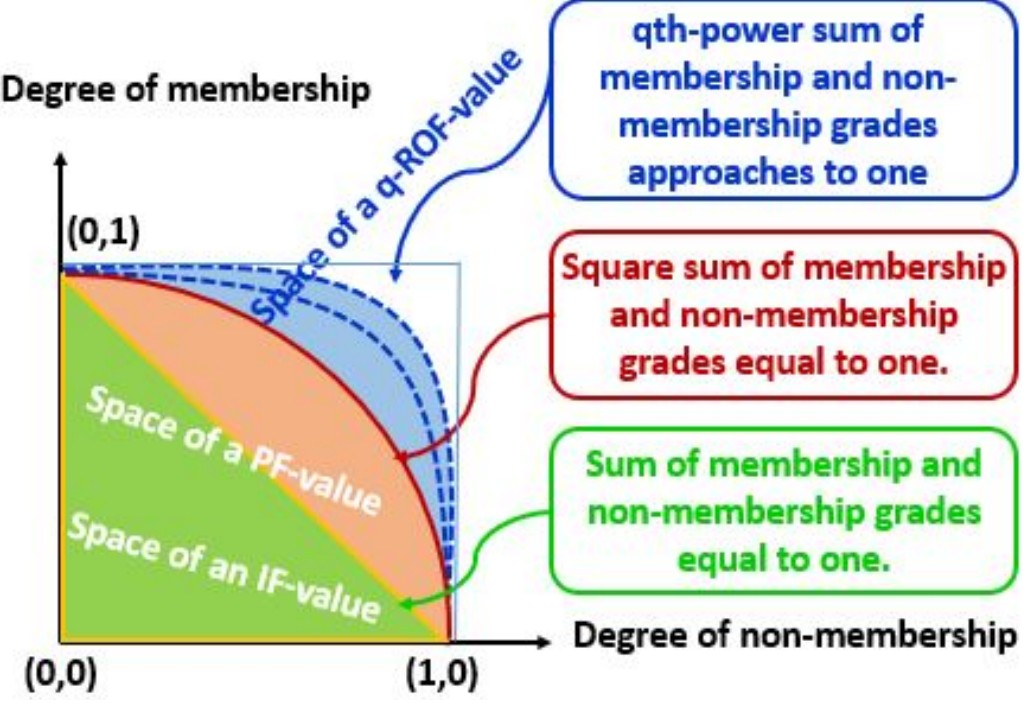

**Figure 1** Geometrical representation of q-ROPFS.

$$\ldots, \alpha^{\aleph}(m_d))$$

$$= \bigoplus_{g=1}^{d} \left( \gamma^{\beta}(m_g) \cdot_{\epsilon} \alpha^{\aleph}(m_g) \right)$$

$$= \gamma^{\beta}(m_1) \cdot_{\epsilon} \alpha^{\aleph}(m_1) \oplus_{\epsilon}, \ldots, \oplus_{\epsilon} \gamma^{\beta}(m_d) \cdot_{\epsilon} \alpha^{\aleph}(m_d)$$

then DQROPFEWA is called "dynamic q-rung orthopair fuzzy Einstein weighted averaging (DQROPFEWA) operator".

**Theorem 3.2** Let $\alpha^{\aleph}(m_k) = \left( \mu^{\zeta}{}_{\alpha^{\aleph}(m_k)}, \nu^{\zeta}{}_{\alpha^{\aleph}(m_k)} \right) (k = 1, \ldots, d)$ be the assortment of q-ROPF values for $d$ distinct time periods) $(k = 1, 2, \ldots, d)$. We can also find the DQROPFEWA operator by,

$$DQROPFEWA\left( \alpha^{\aleph}(m_1), \alpha^{\aleph}(m_2), \ldots, \alpha^{\aleph}(m_d) \right) =$$

$$\left( \sqrt[q]{ \frac{\prod_{g=1}^{d} \left( 1 + \mu^{\zeta}{}_{\alpha^{\aleph}(m_g)}^{q} \right)^{\gamma^{\beta}(m_g)} - \prod_{g=1}^{d} \left( 1 - \mu^{\zeta}{}_{\alpha^{\aleph}(m_g)}^{q} \right)^{\gamma^{\beta}(m_g)}}{\prod_{g=1}^{d} \left( 1 + \mu^{\zeta}{}_{\alpha^{\aleph}(m_g)}^{q} \right)^{\gamma^{\beta}(m_g)} + \prod_{g=1}^{d} \left( 1 - \mu^{\zeta}{}_{\alpha^{\aleph}(m_g)}^{q} \right)^{\gamma^{\beta}(m_g)}} }, \right.$$

$$\left. \frac{\sqrt[q]{2} \prod_{g=1}^{d} \left( \nu^{\zeta}{}_{\alpha^{\aleph}(m_g)} \right)^{\gamma^{\beta}(m_g)}}{\sqrt[q]{\prod_{g=1}^{d} \left( 2 - \nu^{\zeta}{}_{\alpha^{\aleph}(m_g)}^{q} \right)^{\gamma^{\beta}(m_g)} + \prod_{g=1}^{d} \left( \nu^{\zeta}{}_{\alpha^{\aleph}(m_g)}^{q} \right)^{\gamma^{\beta}(m_g)}}} \right)$$

Here, $\gamma^\beta(k) = \left[\gamma^\beta(m_1), \gamma^\beta(m_2), \ldots, \gamma^\beta(m_d)\right]^T$ is the WV of the $d$ distinct time periods and $\sum_{k=1}^{d} \gamma^\beta(m_k) = 1$.

This theorem is proven using mathematical induction.

For $g = 2$

$DQROPFEWA(\alpha^\aleph(m_1), \alpha^\aleph(m_2)) = \gamma^\beta(m_1) \cdot_\epsilon \alpha^\aleph(m_1) \oplus_\epsilon \gamma^\beta(m_2) \cdot_\epsilon \alpha^\aleph(m_2)$

As we know that both $\gamma^\beta(m_1) \cdot_\epsilon \alpha^\aleph(m_1)$ and $\gamma^\beta(m_2) \cdot_\epsilon \alpha^\aleph(m_2)$ are q-ROPFNs, and also $\gamma^\beta(m_1) \cdot_\epsilon \alpha^\aleph(m_1) \oplus_\epsilon \gamma^\beta(m_2) \cdot_\epsilon \alpha^\aleph(m_2)$ is q-ROPFN.

$$\gamma^\beta(m_1) \cdot_\epsilon \alpha^\aleph(m_1) = \left(\sqrt[q]{\frac{(1+\mu\zeta^q_{\alpha^\aleph(m_1)})^{\gamma^\beta(m_1)} - (1-\mu\zeta^q_{\alpha^\aleph(m_1)})^{\gamma^\beta(m_1)}}{(1+\mu\zeta^q_{\alpha^\aleph(m_1)})^{\gamma^\beta(m_1)} + (1-\mu\zeta^q_{\alpha^\aleph(m_1)})^{\gamma^\beta(m_1)}}},\right.$$
$$\left.\frac{\sqrt[q]{2}(\nu\zeta_1)^{\gamma^\beta(m_1)}}{\sqrt[q]{(2-\nu\zeta^q_{\alpha^\aleph(m_1)})^{\gamma^\beta(m_1)} + (\nu\zeta^q_{\alpha^\aleph(m_1)})^{\gamma^\beta(m_1)}}}\right)$$

$$\gamma^\beta(m_2) \cdot_\epsilon \alpha^\aleph(m_2) = \left(\sqrt[q]{\frac{(1+\mu\zeta^q_{\alpha^\aleph(m_2)})^{\gamma^\beta(m_2)} - (1-\mu\zeta^q_{\alpha^\aleph(m_2)})^{\gamma^\beta(m_2)}}{(1+\mu\zeta^q_{\alpha^\aleph(m_2)})^{\gamma^\beta(m_2)} + (1-\mu\zeta^q_{\alpha^\aleph(m_2)})^{\gamma^\beta(m_2)}}},\right.$$
$$\left.\frac{\sqrt[q]{2}(\nu\zeta_2)^{\gamma^\beta(m_2)}}{\sqrt[q]{(2-\nu\zeta^q_{\alpha^\aleph(m_2)})^{\gamma^\beta(m_2)} + (\nu\zeta^q_{\alpha^\aleph(m_2)})^{\gamma^\beta(m_2)}}}\right)$$

Then
$DQROPFEWA(\alpha_1^\aleph, \alpha_2^\aleph)$

$$= \gamma^\beta(m_1) \cdot_\epsilon \alpha_1^\aleph \oplus_\epsilon \gamma^\beta(m_2) \cdot_\epsilon \alpha_2^\aleph$$
$$= \left(\sqrt[q]{\frac{(1+\mu\zeta^q_{\alpha^\aleph(m_1)})^{\gamma^\beta(m_1)} - (1-\mu\zeta^q_{\alpha^\aleph(m_1)})^{\gamma^\beta(m_1)}}{(1+\mu\zeta^q_{\alpha^\aleph(m_1)})^{\gamma^\beta(m_1)} + (1-\mu\zeta^q_{\alpha^\aleph(m_1)})^{\gamma^\beta(m_1)}}}, \frac{\sqrt[q]{2}(\nu\zeta_1)^{\gamma^\beta(m_1)}}{\sqrt[q]{(2-\nu\zeta^q_{\alpha^\aleph(m_1)})^{\gamma^\beta(m_1)} + (\nu\zeta^q_{\alpha^\aleph(m_1)})^{\gamma^\beta(m_1)}}}\right)$$
$$\oplus_\epsilon \left(\sqrt[q]{\frac{(1+\mu\zeta^q_{\alpha^\aleph(m_2)})^{\gamma^\beta(m_2)} - (1-\mu\zeta^q_{\alpha^\aleph(m_2)})^{\gamma^\beta(m_2)}}{(1+\mu\zeta^q_{\alpha^\aleph(m_2)})^{\gamma^\beta(m_2)} + (1-\mu\zeta^q_{\alpha^\aleph(m_2)})^{\gamma^\beta(m_2)}}}, \frac{\sqrt[q]{2}(\nu\zeta_2)^{\gamma^\beta(m_2)}}{\sqrt[q]{(2-\nu\zeta^q_{\alpha^\aleph(m_2)})^{\gamma^\beta(m_2)} + (\nu\zeta^q_{\alpha^\aleph(m_2)})^{\gamma^\beta(m_2)}}}\right)$$
$$= \left(\sqrt[q]{\frac{\frac{(1+\mu\zeta^q_{\alpha^\aleph(m_1)})^{\gamma^\beta(m_1)} - (1-\mu\zeta^q_{\alpha^\aleph(m_1)})^{\gamma^\beta(m_1)}}{(1+\mu\zeta^q_{\alpha^\aleph(m_1)})^{\gamma^\beta(m_1)} + (1-\mu\zeta^q_{\alpha^\aleph(m_1)})^{\gamma^\beta(m_1)}} + \frac{(1+\mu\zeta^q_{\alpha^\aleph(m_2)})^{\gamma^\beta(m_2)} - (1-\mu\zeta^q_{\alpha^\aleph(m_2)})^{\gamma^\beta(m_2)}}{(1+\mu\zeta^q_{\alpha^\aleph(m_2)})^{\gamma^\beta(m_2)} + (1-\mu\zeta^q_{\alpha^\aleph(m_2)})^{\gamma^\beta(m_2)}}}{1 + \left(\frac{(1+\mu\zeta^q_{\alpha^\aleph(m_1)})^{\gamma^\beta(m_1)} - (1-\mu\zeta^q_{\alpha^\aleph(m_1)})^{\gamma^\beta(m_1)}}{(1+\mu\zeta^q_{\alpha^\aleph(m_1)})^{\gamma^\beta(m_1)} + (1-\mu\zeta^q_{\alpha^\aleph(m_1)})^{\gamma^\beta(m_1)}}\right) \cdot_\epsilon \left(\frac{(1+\mu\zeta^q_{\alpha^\aleph(m_2)})^{\gamma^\beta(m_2)} - (1-\mu\zeta^q_{\alpha^\aleph(m_2)})^{\gamma^\beta(m_2)}}{(1+\mu\zeta^q_{\alpha^\aleph(m_2)})^{\gamma^\beta(m_2)} + (1-\mu\zeta^q_{\alpha^\aleph(m_2)})^{\gamma^\beta(m_2)}}\right)}},\right.$$
$$\left.\frac{\left(\frac{\sqrt[q]{2}(\nu\zeta_1)^{\gamma^\beta(m_1)}}{\sqrt[q]{(2-\mu\zeta^q_{\alpha^\aleph(m_1)})^{\gamma^\beta(m_1)} + (\nu\zeta^q_{\alpha^\aleph(m_1)})^{\gamma^\beta(m_1)}}}\right) \cdot_\epsilon \left(\frac{\sqrt[q]{2}(\nu\zeta_2)^{\gamma^\beta(m_2)}}{\sqrt[q]{(2-\mu\zeta^q_{\alpha^\aleph(m_2)})^{\gamma^\beta(m_2)} + (\nu\zeta^q_{\alpha^\aleph(m_2)})^{\gamma^\beta(m_2)}}}\right)}{\sqrt[q]{1 + \left(1 - \frac{2(\nu\zeta^q_{\alpha^\aleph(m_1)})^{\gamma^\beta(m_1)}}{(2-\nu\zeta^q_{\alpha^\aleph(m_1)})^{\gamma^\beta(m_1)} + (\nu\zeta^q_{\alpha^\aleph(m_1)})^{\gamma^\beta(m_1)}}\right) \cdot_\epsilon \left(1 - \frac{2(\nu\zeta^q_{\alpha^\aleph(m_2)})^{\gamma^\beta(m_2)}}{(2-\nu\zeta^q_{\alpha^\aleph(m_2)})^{\gamma^\beta(m_2)} + (\nu\zeta^q_{\alpha^\aleph(m_2)})^{\gamma^\beta(m_2)}}\right)}}\right)$$

$$= \left( \sqrt[q]{\frac{(1+\mu^{\zeta}{}_{\alpha^{\aleph}(m_1)}^{q})^{\gamma^{\beta}(m_1)} \cdot_{\epsilon} (1+\mu^{\zeta}{}_{\alpha^{\aleph}(m_2)}^{q})^{\gamma^{\beta}(m_2)} - (1-\mu^{\zeta}{}_{\alpha^{\aleph}(m_1)}^{q})^{\gamma^{\beta}(m_1)} \cdot_{\epsilon} (1-\mu^{\zeta}{}_{\alpha^{\aleph}(m_2)}^{q})^{\gamma^{\beta}(m_2)}}{(1+\mu^{\zeta}{}_{\alpha^{\aleph}(m_1)}^{q})^{\gamma^{\beta}(m_1)} \cdot_{\epsilon} (1+\mu^{\zeta}{}_{\alpha^{\aleph}(m_2)}^{q})^{\gamma^{\beta}(m_2)} + (1-\mu^{\zeta}{}_{\alpha^{\aleph}(m_1)}^{q})^{\gamma^{\beta}(m_1)} \cdot_{\epsilon} (1-\mu^{\zeta}{}_{\alpha^{\aleph}(m_2)}^{q})^{\gamma^{\beta}(m_2)}}},\right.$$

$$\left. \frac{\sqrt[q]{2}(\nu^{\zeta}{}_{1}^{\gamma^{\beta}(m_1)} \nu^{\zeta}{}_{2}^{\gamma^{\beta}(m_2)})}{\sqrt[q]{(2-\nu^{\zeta}{}_{\alpha^{\aleph}(m_1)}^{q})^{\gamma^{\beta}(m_1)} \cdot_{\epsilon} (2-\nu^{\zeta}{}_{\alpha^{\aleph}(m_2)}^{q})^{\gamma^{\beta}(m_2)} + (\nu^{\zeta}{}_{\alpha^{\aleph}(m_1)}^{q})^{\gamma^{\beta}(m_1)} \cdot_{\epsilon} (\nu^{\zeta}{}_{\alpha^{\aleph}(m_2)}^{q})^{\gamma^{\beta}(m_2)}}} \right)$$

$$= \left( \sqrt[q]{\frac{\prod_{g=1}^{2}\left(1+\mu^{\zeta}{}_{\alpha^{\aleph}(m_g)}^{q}\right)^{\gamma^{\beta}(m_g)} - \prod_{g=1}^{2}\left(1-\mu^{\zeta}{}_{\alpha^{\aleph}(m_g)}^{q}\right)^{\gamma^{\beta}(m_g)}}{\prod_{g=1}^{2}\left(1+\mu^{\zeta}{}_{\alpha^{\aleph}(m_g)}^{q}\right)^{\gamma^{\beta}(m_g)} + \prod_{g=1}^{2}\left(1-\mu^{\zeta}{}_{\alpha^{\aleph}(m_g)}^{q}\right)^{\gamma^{\beta}(m_g)}}},\right.$$

$$\left. \frac{\sqrt[q]{2}\prod_{g=1}^{2}\left(\nu^{\zeta}{}_{\alpha^{\aleph}(m_g)}\right)^{\gamma^{\beta}(m_g)}}{\sqrt[q]{\prod_{g=1}^{2}\left(2-\nu^{\zeta}{}_{\alpha^{\aleph}(m_g)}^{q}\right)^{\gamma^{\beta}(m_g)} + \prod_{g=1}^{2}\left(\nu^{\zeta}{}_{\alpha^{\aleph}(m_g)}^{q}\right)^{\gamma^{\beta}(m_g)}}} \right)$$

which proves for $g = 2$.

Assume that result is true for $g = r$, we have

$DQROPFEWA\left(\alpha^{\aleph}(m_1), \alpha^{\aleph}(m_2), \ldots, \alpha^{\aleph}(m_r)\right)$

$$= \left( \sqrt[q]{\frac{\prod_{g=1}^{r}\left(1+\mu^{\zeta}{}_{\alpha^{\aleph}(m_g)}^{q}\right)^{\gamma^{\beta}(m_g)} - \prod_{g=1}^{r}\left(1-\mu^{\zeta}{}_{\alpha^{\aleph}(m_g)}^{q}\right)^{\gamma^{\beta}(m_g)}}{\prod_{g=1}^{r}\left(1+\mu^{\zeta}{}_{\alpha^{\aleph}(m_g)}^{q}\right)^{\gamma^{\beta}(m_g)} + \prod_{g=1}^{r}\left(1-\mu^{\zeta}{}_{\alpha^{\aleph}(m_g)}^{q}\right)^{\gamma^{\beta}(m_g)}}},\right.$$

$$\left. \frac{\sqrt[q]{2}\prod_{g=1}^{r}\left(\nu^{\zeta}{}_{\alpha^{\aleph}(m_g)}\right)^{\gamma^{\beta}(m_g)}}{\sqrt[q]{\prod_{g=1}^{r}\left(2-\nu^{\zeta}{}_{\alpha^{\aleph}(m_g)}^{q}\right)^{\gamma^{\beta}(m_g)} + \prod_{g=1}^{r}\left(\nu^{\zeta}{}_{\alpha^{\aleph}(m_g)}^{q}\right)^{\gamma^{\beta}(m_g)}}} \right)$$

Now we will prove for $g = r+1$,

$DQROPFEWA\left(\alpha^{\aleph}(m_1), \alpha^{\aleph}(m_2), \ldots, \alpha^{\aleph}(m_{r+1})\right)$

$= DQROPFEWA\left(\alpha^{\aleph}(m_1), \alpha^{\aleph}(m_2), \ldots, \alpha^{\aleph}(m_r)\right) \oplus \gamma^{\beta}(m_{r+1}) \cdot_{\epsilon} \alpha^{\aleph}(m_{r+1})$

$$= \left( \sqrt[q]{\frac{\prod_{g=1}^{r}\left(1+\mu^{\zeta}{}_{\alpha^{\aleph}(m_g)}^{q}\right)^{\gamma^{\beta}(m_g)} - \prod_{g=1}^{r}\left(1-\mu^{\zeta}{}_{\alpha^{\aleph}(m_g)}^{q}\right)^{\gamma^{\beta}(m_g)}}{\prod_{g=1}^{r}\left(1+\mu^{\zeta}{}_{\alpha^{\aleph}(m_g)}^{q}\right)^{\gamma^{\beta}(m_g)} + \prod_{g=1}^{r}\left(1-\mu^{\zeta}{}_{\alpha^{\aleph}(m_g)}^{q}\right)^{\gamma^{\beta}(m_g)}}},\right.$$

$$\left. \frac{\sqrt[q]{2}\prod_{g=1}^{r}\left(\nu^{\zeta}{}_{\alpha^{\aleph}(m_g)}\right)^{\gamma^{\beta}(m_g)}}{\sqrt[q]{\prod_{g=1}^{r}\left(2-\nu^{\zeta}{}_{\alpha^{\aleph}(m_g)}^{q}\right)^{\gamma^{\beta}(m_g)} + \prod_{g=1}^{r}\left(\nu^{\zeta}{}_{\alpha^{\aleph}(m_g)}^{q}\right)^{\gamma^{\beta}(m_g)}}} \right)$$

$$\oplus \left( \sqrt[q]{\frac{\left(1+\mu^{\zeta}{}_{\alpha^{\aleph}(m_{r+1})}^{q}\right)^{\gamma^{\beta}(m_{r+1})} - \left(1-\mu^{\zeta}{}_{\alpha^{\aleph}(m_{r+1})}^{q}\right)^{\gamma^{\beta}(m_{r+1})}}{\left(1+\mu^{\zeta}{}_{\alpha^{\aleph}(m_{r+1})}^{q}\right)^{\gamma^{\beta}(m_{r+1})} + \left(1-\mu^{\zeta}{}_{\alpha^{\aleph}(m_{r+1})}^{q}\right)^{\gamma^{\beta}(m_{r+1})}}}, \right.$$

$$\left. \frac{\sqrt[q]{2}\left(\nu^{\zeta}_{\alpha^{\aleph}(m_{r+1})}\right)^{\gamma^{\beta}(m_{r+1})}}{\sqrt[q]{\left(2-\nu^{\zeta\,q}_{\alpha^{\aleph}(m_{r+1})}\right)^{\gamma^{\beta}(m_{r+1})}+\left(\nu^{\zeta\,q}_{\alpha^{\aleph}(m_{r+1})}\right)^{\gamma^{\beta}(m_{r+1})}}} \right)$$

$$=\left(\sqrt[q]{\frac{\prod_{g=1}^{r+1}\left(1+\mu^{\zeta\,q}_{\alpha^{\aleph}(m_g)}\right)^{\gamma^{\beta}(m_g)}-\prod_{g=1}^{r+1}\left(1-\mu^{\zeta\,q}_{\alpha^{\aleph}(m_g)}\right)^{\gamma^{\beta}(m_g)}}{\prod_{g=1}^{r+1}\left(1+\mu^{\zeta\,q}_{\alpha^{\aleph}(m_g)}\right)^{\gamma^{\beta}(m_g)}+\prod_{g=1}^{r+1}\left(1-\mu^{\zeta\,q}_{\alpha^{\aleph}(m_g)}\right)^{\gamma^{\beta}(m_g)}}},\right.$$

$$\left. \frac{\sqrt[q]{2}\prod_{g=1}^{r+1}\left(\nu^{\zeta}_{\alpha^{\aleph}(m_g)}\right)^{\gamma^{\beta}(m_g)}}{\sqrt[q]{\prod_{g=1}^{r+1}\left(2-\nu^{\zeta\,q}_{\alpha^{\aleph}(m_g)}\right)^{\gamma^{\beta}(m_g)}+\prod_{g=1}^{r+1}\left(\nu^{\zeta\,q}_{\alpha^{\aleph}(m_g)}\right)^{\gamma^{\beta}(m_g)}}} \right)$$

thus the result holds for $g = r+1$. This proves the required result.

**Theorem 3.3** *Let* $\alpha^{\aleph}(m_k) = \left(\mu^{\zeta}_{\alpha^{\aleph}(m_k)}, \nu^{\zeta}_{\alpha^{\aleph}(m_k)}\right)$ *be the family of q-ROPFNs. Aggregated value using DQROPFEWA operator is q-ROPFN.*

Suppose $\alpha^{\aleph}(m_k) = \left(\mu^{\zeta}_{\alpha^{\aleph}(m_k)}, \nu^{\zeta}_{\alpha^{\aleph}(m_k)}\right)$ is the family of q-ROPFNs. By definition of q-ROPFN,

$$0 \le \mu^{\zeta\,q}_{\alpha^{\aleph}(m_k)} + \nu^{\zeta\,q}_{\alpha^{\aleph}(m_k)} \le 1.$$

Therefore,

$$\frac{\prod_{k=1}^{d} 1+(\mu^{\zeta\,q}_{\alpha^{\aleph}(m_k)})^{\gamma^{\beta}(m_k)}-\prod_{k=1}^{d}(1-(\mu^{\zeta\,q}_{\alpha^{\aleph}(m_k)})^q)^{\gamma^{\beta}(m_k)}}{\prod_{k=1}^{d}(1+(\mu^{\zeta\,q}_{\alpha^{\aleph}(m_k)})^q)^{\gamma^{\beta}(m_k)}+\prod_{k=1}^{d}(1-(\mu^{\zeta\,q}_{\alpha^{\aleph}(m_k)})^q)^{\gamma^{\beta}(m_k)}}$$

$$=1-\frac{2\prod_{k=1}^{d}(1-(\mu^{\zeta\,q}_{\alpha^{\aleph}(m_k)})^q)^{\gamma^{\beta}(m_k)}}{\prod_{k=1}^{d}(1+(\mu^{\zeta\,q}_{\alpha^{\aleph}(m_k)})^q)^{\gamma^{\beta}(m_k)}+\prod_{k=1}^{d}(1-(\mu^{\zeta\,q}_{\alpha^{\aleph}(m_k)})^q)^{\gamma^{\beta}(m_k)}}$$

$$\le 1-\prod_{k=1}^{d}(1-(\mu^{\zeta\,q}_{\alpha^{\aleph}(m_k)})^q)^{\gamma^{\beta}(m_k)} \le 1$$

and

$$(1+(\mu^{\zeta\,q}_{\alpha^{\aleph}(m_k)})^q)^{\gamma^{\beta}(m_k)} \ge (1-(\mu^{\zeta\,q}_{\alpha^{\aleph}(m_k)})^q)^{\gamma^{\beta}(m_k)}$$

$$\prod_{k=1}^{d}(1+(\mu^{\zeta\,q}_{\alpha^{\aleph}(m_k)})^q)^{\gamma^{\beta}(m_k)} \ge \prod_{k=1}^{d}(1-(\mu^{\zeta\,q}_{\alpha^{\aleph}(m_k)})^q)^{\gamma^{\beta}(m_k)}$$

$$\prod_{k=1}^{d}(1+(\mu^{\zeta\,q}_{\alpha^{\aleph}(m_k)})^q)^{\gamma^{\beta}(m_k)} - \prod_{k=1}^{d}(1-(\mu^{\zeta\,q}_{\alpha^{\aleph}(m_k)})^q)^{\gamma^{\beta}(m_k)} \ge 0$$

$$\frac{\prod_{k=1}^{d}(1+(\mu^{\zeta\,q}_{\alpha^{\aleph}(m_k)})^q)^{\gamma^{\beta}(m_k)} - \prod_{k=1}^{d}(1-(\mu^{\zeta\,q}_{\alpha^{\aleph}(m_k)})^q)^{\gamma^{\beta}(m_k)}}{\prod_{k=1}^{d}(1+(\mu^{\zeta\,q}_{\alpha^{\aleph}(m_k)})^q)^{\gamma^{\beta}(m_k)} + \prod_{k=1}^{d}(1-(\mu^{\zeta\,q}_{\alpha^{\aleph}(m_k)})^q)^{\gamma^{\beta}(m_k)}} \ge 0$$

So, we get $0 \le \mu^{\zeta}_{\text{DQROPFEWA}} \le 1$. For $\nu^{\zeta}_{\text{DQROPFEWA}}$, we have

$$\frac{2\prod_{k=1}^{d}(\nu^{\zeta\,q}_{\alpha^{\aleph}(m_k)})^{\gamma^{\beta}(m_k)}}{\prod_{k=1}^{d}(1+(\mu^{\zeta\,q}_{\alpha^{\aleph}(m_k)})^q)^{\gamma^{\beta}(m_k)} + \prod_{k=1}^{d}(1-(\mu^{\zeta\,q}_{\alpha^{\aleph}(m_k)})^q)^{\gamma^{\beta}(m_k)}}$$

$$\leq \frac{2\prod_{k=1}^{d}(1-(\mu^{\zeta\,q}_{\alpha^{\aleph}(m_k)})^q)^{\gamma^{\beta}(m_k)}}{\prod_{k=1}^{d}(1+(\mu^{\zeta\,q}_{\alpha^{\aleph}(m_k)})^q)^{\gamma^{\beta}(m_k)}+\prod_{k=1}^{d}(1-(\mu^{\zeta\,q}_{\alpha^{\aleph}(m_k)})^q)^{\gamma^{\beta}(m_k)}}$$

$$\leq \prod_{k=1}^{d}(1-(\mu^{\zeta\,q}_{\alpha^{\aleph}(m_k)})^q)^{\gamma^{\beta}(m_k)}$$

$$\leq 1$$

Also,

$$\frac{2\prod_{k=1}^{d}(\nu^{\zeta\,q}_{\alpha^{\aleph}(m_k)})^{\gamma^{\beta}(m_k)}}{\prod_{k=1}^{d}(1+(\mu^{\zeta\,q}_{\alpha^{\aleph}(m_k)})^q)^{\gamma^{\beta}(m_k)}+\prod_{k=1}^{d}(1-(\mu^{\zeta\,q}_{\alpha^{\aleph}(m_k)})^q)^{\gamma^{\beta}(m_k)}}\geq 0 \Longleftrightarrow \prod_{k=1}^{d}(\nu^{\zeta\,q}_{\alpha^{\aleph}(m_k)})^{\gamma^{\beta}(m_k)}\geq 0$$

Moreover,

$$\mu^{\zeta\,q}_{\text{DQROPFEWA}}+\nu^{\zeta\,q}_{\text{DQROPFEWA}} = \frac{\prod_{k=1}^{d}(1+(\mu^{\zeta\,q}_{\alpha^{\aleph}(m_k)})^q)^{\gamma^{\beta}(m_k)}-\prod_{k=1}^{d}(1-(\mu^{\zeta\,q}_{\alpha^{\aleph}(m_k)})^q)^{\gamma^{\beta}(m_k)}}{\prod_{k=1}^{d}(1+(\mu^{\zeta\,q}_{\alpha^{\aleph}(m_k)})^q)^{\gamma^{\beta}(m_k)}+\prod_{k=1}^{d}(1-(\mu^{\zeta\,q}_{\alpha^{\aleph}(m_k)})^q)^{\gamma^{\beta}(m_k)}}+$$

$$\frac{2\prod_{k=1}^{d}(\nu^{\zeta\,q}_{\alpha^{\aleph}(m_k)})^{\gamma^{\beta}(m_k)}}{\prod_{k=1}^{d}(2-\nu^{\zeta\,q}_{\alpha^{\aleph}(m_k)})^{\gamma^{\beta}(m_k)}+\prod_{k=1}^{d}(\nu^{\zeta\,q}_{\alpha^{\aleph}(m_k)})^{\gamma^{\beta}(m_k)}}$$

$$\leq 1 - \frac{2\prod_{k=1}^{d}(1-(\mu^{\zeta\,q}_{\alpha^{\aleph}(m_k)})^q)^{\gamma^{\beta}(m_k)}}{\prod_{k=1}^{d}(1+(\mu^{\zeta\,q}_{\alpha^{\aleph}(m_k)})^q)^{\gamma^{\beta}(m_k)}+\prod_{k=1}^{d}(1-(\mu^{\zeta\,q}_{\alpha^{\aleph}(m_k)})^q)^{\gamma^{\beta}(m_k)}}+$$

$$\frac{2\prod_{k=1}^{d}(1-(\mu^{\zeta\,q}_{\alpha^{\aleph}(m_k)})^q)^{\gamma^{\beta}(m_k)}}{\prod_{k=1}^{d}(1+(\mu^{\zeta\,q}_{\alpha^{\aleph}(m_k)})^q)^{\gamma^{\beta}(m_k)}+\prod_{k=1}^{d}(1-(\mu^{\zeta\,q}_{\alpha^{\aleph}(m_k)})^q)^{\gamma^{\beta}(m_k)}}$$

$$\leq 1$$

Thus, DQROPFEWA $\in [0,1]$. Consequently, q-ROPFNs gathered by the DQROPFEWA operator also are q-ROPFNs. We can easily show that following properties.

**Theorem 3.4** *Let* $\alpha^{\aleph}(m_k)=\left(\mu^{\zeta}_{\alpha^{\aleph}(m_k)},\nu^{\zeta}_{\alpha^{\aleph}(m_k)}\right)$ *$(k=1,\ldots,d)$ be the assortment of q-ROPF values for d distinct time periods $(k=1,2,\ldots,d)$ and all* $\alpha^{\aleph}(m_k)=\left(\mu^{\zeta}_{\alpha^{\aleph}(m_k)},\nu^{\zeta}_{\alpha^{\aleph}(m_k)}\right)$ *$(k=1,\ldots,d)$ are equal, i.e.,* $\alpha^{\aleph}(m_k)=\alpha^{\aleph}$ *for all k, then*
$DQROPFEWA\left(\alpha^{\aleph}(m_1),\alpha^{\aleph}(m_2),\ldots,\alpha^{\aleph}(m_d)\right)=\alpha^{\aleph}.$

Since $\alpha^{\aleph}(m_k)=\alpha^{\aleph}$, for all $k=1,\ldots,p$, i.e., $\mu^{\zeta}_{\alpha^{\aleph}(m_k)}=\mu^{\zeta}_{\alpha^{\aleph}}$ and $\nu^{\zeta}_{\alpha^{\aleph}(m_k)}=\nu^{\zeta}_{\alpha^{\aleph}}, k=1,\ldots,p$, then

$$DQROPFEWA\left(\alpha^{\aleph}(m_1),\alpha^{\aleph}(m_2),\ldots,\alpha^{\aleph}(m_d)\right)$$

$$=\left(\frac{\prod_{g=1}^{d}\left(1+\mu^{\zeta}_{\alpha^{\aleph}(m_g)}\right)^{\gamma^{\beta}(m_g)}-\prod_{g=1}^{d}\left(1-\mu^{\zeta}_{\alpha^{\aleph}(m_g)}\right)^{\gamma^{\beta}(m_g)}}{\prod_{g=1}^{d}\left(1+\mu^{\zeta}_{\alpha^{\aleph}(m_g)}\right)^{\gamma^{\beta}(m_g)}+\prod_{g=1}^{d}\left(1-\mu^{\zeta}_{\alpha^{\aleph}(m_g)}\right)^{\gamma^{\beta}(m_g)}},\right.$$

$$\left.\frac{2\prod_{g=1}^{d}\left(\nu^{\zeta}_{\alpha^{\aleph}(m_g)}\right)^{\gamma^{\beta}(m_g)}}{\prod_{g=1}^{d}\left(2-\nu^{\zeta}_{\alpha^{\aleph}(m_g)}\right)^{\gamma^{\beta}(m_g)}+\prod_{g=1}^{d}\left(\nu^{\zeta}_{\alpha^{\aleph}(m_g)}\right)^{\gamma^{\beta}(m_g)}}\right)$$

$$=\left(\frac{\left(1+\mu^{\zeta}_{\alpha^{\aleph}(m_g)}\right)^{\sum_{g=1}^{d}\gamma^{\beta}(m_g)}-\left(1-\mu^{\zeta}_{\alpha^{\aleph}(m_g)}\right)^{\sum_{g=1}^{d}\gamma^{\beta}(m_g)}}{\left(1+\mu^{\zeta}_{\alpha^{\aleph}(m_g)}\right)^{\sum_{g=1}^{d}\gamma^{\beta}(m_g)}+\left(1-\mu^{\zeta}_{\alpha^{\aleph}(m_g)}\right)^{\sum_{g=1}^{d}\gamma^{\beta}(m_g)}},\right.$$

$$\frac{2\left(\nu^{\zeta}{}_{\alpha^{\aleph}(m_g)}\right)^{\sum_{g=1}^{d}\gamma^{\beta}(m_g)}}{\left(2-\nu^{\zeta}{}_{\alpha^{\aleph}(m_g)}\right)^{\sum_{g=1}^{d}\gamma^{\beta}(m_g)}+\left(\nu^{\zeta}{}_{\alpha^{\aleph}(m_g)}\right)^{\sum_{g=1}^{d}\gamma^{\beta}(m_g)}}\Bigg)$$

$$=\left(\frac{\left(1+\mu^{\zeta}{}_{\alpha^{\aleph}(m_g)}\right)-\left(1-\mu^{\zeta}{}_{\alpha^{\aleph}(m_g)}\right)}{\left(1+\mu^{\zeta}{}_{\alpha^{\aleph}(m_g)}\right)+\left(1-\mu^{\zeta}{}_{\alpha^{\aleph}(m_g)}\right)},\frac{2\left(\nu^{\zeta}{}_{\alpha^{\aleph}(m_g)}\right)}{\left(2-\nu^{\zeta}{}_{\alpha^{\aleph}(m_g)}\right)+\left(\nu^{\zeta}{}_{\alpha^{\aleph}(m_g)}\right)}\right)$$

$$=\left(\mu^{\zeta}{}_{\alpha^{\aleph}(m_g)},\nu^{\zeta}{}_{\alpha^{\aleph}(m_g)}\right)=\alpha^{\aleph}.$$

**Theorem 3.5** *Assume that* $\alpha^{\aleph}(m_k)=\langle\mu^{\zeta}{}_{\alpha^{\aleph}(m_k)},\nu^{\zeta}{}_{\alpha^{\aleph}(m_k)}\rangle$ *be the family of q-ROPFNs, then*

$$\alpha^{\aleph}_{min}\leq DQROPFEWA\left(\alpha^{\aleph}(m_1),\alpha^{\aleph}(m_2),\dots,\alpha^{\aleph}(m_d)\right) \tag{2}$$

$$\leq\alpha^{\aleph}_{max}$$
*where,*

$$\alpha^{\aleph}_{min}=min(\alpha^{\aleph}(m_k)),\quad\alpha^{\aleph}_{max}=max(\alpha^{\aleph}(m_k))$$

**Theorem 3.6** *(Monotonicity) Assume that* $\alpha^{\aleph}(m_k)=\langle\mu^{\zeta}{}_{\alpha^{\aleph}(m_k)},\nu^{\zeta}{}_{\alpha^{\aleph}(m_k)}\rangle$ *and* $\alpha^{\aleph*}(m_k)=\langle\mu^{\zeta*}{}_{\alpha^{\aleph}(m_k)},\nu^{\zeta*}{}_{\alpha^{\aleph}(m_k)}\rangle$ *are the families of q-ROPFNs. If* $\mu^{\zeta*}{}_{\alpha^{\aleph}(m_k)}\geq\mu^{\zeta}{}_{\alpha^{\aleph}(m_k)}$ *and* $\nu^{\zeta*}{}_{\alpha^{\aleph}(m_k)}\leq\nu^{\zeta}{}_{\alpha^{\aleph}(m_k)}$ *for all* $j,$ *then*

$$DQROPFEWA\left(\alpha^{\aleph}(m_1),\alpha^{\aleph}(m_2),\dots,\alpha^{\aleph}(m_d)\right)\leq$$
$$DQROPFEWA\left(\alpha^{\aleph*}(m_1),\alpha^{\aleph*}(m_2),\dots,\alpha^{\aleph*}(m_d)\right)$$

## DQROPFEWG operator

**Definition 3.7** *Let* $\alpha^{\aleph}(m_k)=\left(\mu^{\zeta}{}_{\alpha^{\aleph}(m_k)},\nu^{\zeta}{}_{\alpha^{\aleph}(m_k)}\right)(k=1,\dots,d)$ *be the assortment of q-ROPF values for d distinct time periods* $(k=1,2,\dots,d).$ $\gamma^{\beta}(k)=\left[\gamma^{\beta}(m_1),\gamma^{\beta}(m_2),\dots,\gamma^{\beta}(m_d)\right]^{T}$ *is the WV of the periods, where* $\sum_{k=1}^{d}\gamma^{\beta}(m_k)=1$ *and let* $DQROPFEWG:X^{n}\to X.$ *If* $DQROPFEWG\left(\alpha^{\aleph}(m_1),\alpha^{\aleph}(m_2),\dots,\alpha^{\aleph}(m_d)\right)$

$$=\bigoplus_{g=1}^{d}\left(\gamma^{\beta}(m_g)\cdot_{\epsilon}\alpha^{\aleph}(m_g)\right)$$
$$=\gamma^{\beta}(m_1)\cdot_{\epsilon}\alpha^{\aleph}(m_1)\oplus_{\epsilon},\dots,\oplus_{\epsilon}\gamma^{\beta}(m_d)\cdot_{\epsilon}\alpha^{\aleph}(m_d)$$

*then DQROPFEWG is called "dynamic q-rung orthopair fuzzy Einstein weighted geometric (DQROPFEWG) operator".*

**Theorem 3.8** *Let* $\alpha^{\aleph}(m_k)=\left(\mu^{\zeta}{}_{\alpha^{\aleph}(m_k)},\nu^{\zeta}{}_{\alpha^{\aleph}(m_k)}\right)(k=1,\dots,d)$ *be the assortment of q-ROPF values for d distinct time periods* $(k=1,2,\dots,d).$ *We can also find the DQROPFEWG operator by,*

$$DQROPFEWG\left(\alpha^{\aleph}(m_1),\alpha^{\aleph}(m_2),\dots,\alpha^{\aleph}(m_d)\right)$$

$$\frac{\sqrt[q]{2}\prod_{g=1}^{d}\left(\mu^{\zeta}{}_{\alpha^{\aleph}(m_g)}\right)^{\gamma^{\beta}(m_g)}}{\sqrt[q]{\prod_{g=1}^{d}\left(2-\mu^{\zeta q}{}_{\alpha^{\aleph}(m_g)}\right)^{\gamma^{\beta}(m_g)}+\prod_{g=1}^{d}\left(\mu^{\zeta q}{}_{\alpha^{\aleph}(m_g)}\right)^{\gamma^{\beta}(m_g)}}}$$

$$\sqrt[q]{\frac{\prod_{g=1}^{d}\left(1+\nu^{\zeta}{}_{\alpha^{\aleph}(m_g)}^{q}\right)^{\gamma^{\beta}(m_g)}-\prod_{g=1}^{d}\left(1-\nu^{\zeta}{}_{\alpha^{\aleph}(m_g)}^{q}\right)^{\gamma^{\beta}(m_g)}}{\prod_{g=1}^{d}\left(1+\nu^{\zeta}{}_{\alpha^{\aleph}(m_g)}^{q}\right)^{\gamma^{\beta}(m_g)}+\prod_{g=1}^{d}\left(1-\nu^{\zeta}{}_{\alpha^{\aleph}(m_g)}^{q}\right)\gamma^{\beta}(mg)}},$$

*Here,* $\gamma^{\beta}(k)=\left[\gamma^{\beta}(m_1),\gamma^{\beta}(m_2),\ldots,\gamma^{\beta}(m_d)\right]^{T}$ *is the WV of the* $d$ *distinct time periods and* $\sum_{k=1}^{d}\gamma^{\beta}(m_k)=1$.

This is same as Theorem 3.2.

**Theorem 3.9** *Let* $\alpha^{\aleph}(m_k)=\left(\mu^{\zeta}{}_{\alpha^{\aleph}(m_k)},\nu^{\zeta}{}_{\alpha^{\aleph}(m_k)}\right)$ *be the family of q-ROPFNs. Aggregated value using DQROPFEWG operator is q-ROPFN.*

This is same as Theorem 3.3.

**Theorem 3.10** *Let* $\alpha^{\aleph}(m_k)=\left(\mu^{\zeta}{}_{\alpha^{\aleph}(m_k)},\nu^{\zeta}{}_{\alpha^{\aleph}(m_k)}\right)$ $(k=1,\ldots,d)$ *be the assortment of q-ROPF values for* $d$ *distinct time periods* $(k=1,2,\ldots,d)$ *and all* $\alpha^{\aleph}(m_k)=\left(\mu^{\zeta}{}_{\alpha^{\aleph}(m_k)},\nu^{\zeta}{}_{\alpha^{\aleph}(m_k)}\right)k=1,\ldots,d$ *are equal, i.e.,* $\alpha^{\aleph}(m_k)=\alpha^{\aleph}$ *for all k, then* $DQROPFEWG\left(\alpha^{\aleph}(m_1),\alpha^{\aleph}(m_2),\ldots,\alpha^{\aleph}(m_d)\right)=\alpha^{\aleph}$.

This is same as Theorem 3.4.

**Theorem 3.11** *Assume that* $\alpha^{\aleph}(m_k)=\langle\mu^{\zeta}{}_{\alpha^{\aleph}(m_k)},\nu^{\zeta}{}_{\alpha^{\aleph}(m_k)}\rangle$ *be the family of q-ROPFNs, then*

$$\alpha^{\aleph}_{min}\leq DQROPFEWG\left(\alpha^{\aleph}(m_1),\alpha^{\aleph}(m_2),\ldots,\alpha^{\aleph}(m_d)\right) \tag{3}$$

$$\leq\alpha^{\aleph}_{max}$$

*where,*

$$\alpha^{\aleph}_{min}=min(\alpha^{\aleph}(m_k)),\quad \alpha^{\aleph}_{max}=max(\alpha^{\aleph}(m_k))$$

**Theorem 3.12** *(Monotonicity) Assume that* $\alpha^{\aleph}(m_k)=\langle\mu^{\zeta}{}_{\alpha^{\aleph}(m_k)},\nu^{\zeta}{}_{\alpha^{\aleph}(m_k)}\rangle$ *and* $\alpha^{\aleph*}(m_k)=\langle\mu^{\zeta*}{}_{\alpha^{\aleph}(m_k)},\nu^{\zeta*}{}_{\alpha^{\aleph}(m_k)}\rangle$ *are the families of q-ROPFNs. If* $\mu^{\zeta*}{}_{\alpha^{\aleph}(m_k)}\geq\mu^{\zeta}{}_{\alpha^{\aleph}(m_k)}$ *and* $\nu^{\zeta*}{}_{\alpha^{\aleph}(m_k)}\leq\nu^{\zeta}{}_{\alpha^{\aleph}(m_k)}$ *italic for all* $j$ *,then*

$$DQROPFEWG\left(\alpha^{\aleph}(m_1),\alpha^{\aleph}(m_2),\ldots,\alpha^{\aleph}(m_d)\right)$$

$$\leq DQROPFEWG\left(\alpha^{\aleph*}(m_1),\alpha^{\aleph*}(m_2),\ldots,\alpha^{\aleph*}(m_d)\right)$$

## MCDM METHODS WITH PROPOSED AOS

Consider $\xi^{\eta}=\left\{\xi_1^{\eta},\xi_2^{\eta},\ldots,\xi_m^{\eta}\right\}$ is the discrete set of $m$ alternatives and $\aleph=\{\aleph_1,\aleph_2,\ldots,\aleph_n\}$ a discrete set of $n$ criteria and whose weights vector is $W=[\Omega_1,\Omega_2,\ldots,\Omega_n]$. $k=1,2,\ldots,d$ is a discrete set of $d$ periods and whose WV is $\gamma^{\beta}(m_k)=\left[\gamma^{\beta}(m_1),\gamma^{\beta}(m_2),\ldots,\gamma^{\beta}(m_d)\right]^{T}$, where $\gamma^{\beta}(m_k)>0$, $\sum_{k=1}^{d}\gamma^{\beta}(m_k)=1$. Let $R(m_k)=\left(r_{ij}^{k}\right)_{m\times n}=\left(\mu^{\zeta'}{}_{ij}(m_k),\nu^{\zeta'}{}_{ij}(m_k)\right)_{m\times n}$ is the decision matrix with q-ROPF values, where $\mu^{\zeta}{}_{ij}(m_k)$ represents the degree that $i^{\text{th}}$ alternative satisfies the $j^{\text{th}}$ criterion at $k^{\text{th}}$ periods, $\nu^{\zeta}{}_{ij}(m_k)$ represents the degree that $i^{\text{th}}$ alternative doesn't satisfy the $j^{\text{th}}$ criterion at $k^{\text{th}}$ periods such that $0\leq\mu^{\zeta'}{}_{ij}(m_k)\leq1$, $0\leq\nu^{\zeta'}{}_{ij}(m_k)\leq1$, $\mu^{\zeta}{}_{ij}^{q}(m_k)+\nu^{\zeta}{}_{ij}^{q}(m_k)\leq1$ for $i=1,2,\ldots,m,j=1,2,\ldots,n,k=1,2,\ldots,p$.

**Algorithm**

---

**Step 1:**

Obtain the decision matrices $R(m_k) = \left(r_{ij}^k\right)_{m \times n} = \left(\mu^{\zeta'}_{ij}(m_k), v^{\zeta'}_{ij}(m_k)\right)_{m \times n}$ for the $d$ distinct time periods.

**Step 2:**

The decision matrix discusses two types of criterion: $(\tau_c)$ cost form key indicators and $(\tau_b)$ benefit form criteria. If all indicators are from the same category, no need for normalisation; nevertheless, in MCDM, there may be two types of parameters. In this scenario, the matrix was updated to the transforming response matrix $N(m_k) = \left(n_{ij}^k\right)_{m \times n} = \left(\mu^{\zeta}_{ij}(m_k), v^{\zeta}_{ij}(m_k)\right)_{m \times n}$ using the normalization formula Eq. (4).

$$\left(n_{ij}^k\right)_{m \times n} = \begin{cases} \left(\left(r_{ij}^k\right)_{m \times n}\right)^c ; j \in \tau_c \\ \left(r_{ij}^k\right)_{m \times n} ; j \in \tau_b. \end{cases} \tag{4}$$

where $\left(\left(r_{ij}^k\right)_{m \times n}\right)^c$ show the compliment of $\left(r_{ij}^k\right)_{m \times n}$.

**Step 3:**

In this step, we utilized one of the suggested AOs to concentration all the "normalized decision matrices" $N(m_k) = \left(n_{ij}^k\right)_{m \times n} = \left(\mu^{\zeta}_{ij}(m_k), v^{\zeta}_{ij}(m_k)\right)_{m \times n}$ into one cumulative q-ROPF matrix $Z = \left(z_{ij}\right)_{m \times n} = \left(\mu^{\zeta}_{ij}, v^{\zeta}_{ij}\right)_{m \times n}$:

$$z_{ij} = DQROPFEWA\left(n_{ij}(m_1), n_{ij}(m_2), \ldots, n_{ij}(m_d)\right)$$

$$= \left(\sqrt[q]{\frac{\prod_{k=1}^{d}\left(1 + \mu^{\zeta\,q}_{n_{ij}(m_k)}\right)^{\gamma^{\beta}(m_k)} - \prod_{k=1}^{d}\left(1 - \mu^{\zeta\,q}_{n_{ij}}(m_k)\right)^{\gamma^{\beta}(m_k)}}{\prod_{k=1}^{d}\left(1 + \mu^{\zeta\,q}_{n_{ij}(m_k)}\right)^{\gamma^{\beta}(m_k)} + \prod_{k=1}^{d}\left(1 - \mu^{\zeta\,q}_{n_{ij}(m_k)}\right)^{\gamma^{\beta}(m_k)}}},\right.$$

$$\left.\frac{\sqrt[q]{2}\prod_{k=1}^{d} v^{\zeta\,\gamma^{\beta}(m_k)}_{n_{ij}(m_k)}}{\sqrt[q]{\prod_{k=1}^{d}\left(2 - v^{\zeta\,q}_{n_{ij}(m_k)}\right)^{\gamma^{\beta}(m_k)} + \prod_{k=1}^{d}\left(v^{\zeta\,q}_{n_{ij}(m_k)}\right)^{\gamma^{\beta}(m_k)}}}\right) \tag{5}$$

or

$$z_{ij} = DQROPFEWG\left(n_{ij}(m_1), n_{ij}(m_2), \ldots, n_{ij}(m_d)\right)$$

$$= \left(\frac{\sqrt[q]{2}\prod_{k=1}^{d} \mu^{\zeta\,\gamma^{\beta}(m_k)}_{n_{ij}(m_k)}}{\sqrt[q]{\prod_{k=1}^{d}\left(2 - \mu^{\zeta\,q}_{n_{ij}(m_k)}\right)^{\gamma^{\beta}(m_k)} + \prod_{k=1}^{d}\left(\mu^{\zeta\,q}_{n_{ij}(m_k)}\right)^{\gamma^{\beta}(m_k)}}},\right.$$

$$\left.\sqrt[q]{\frac{\prod_{k=1}^{d}\left(1 + v^{\zeta\,q}_{n_{ij}(m_k)}\right)^{\gamma^{\beta}(m_k)} - \prod_{k=1}^{d}\left(1 - v^{\zeta\,q}_{n_{ij}}(m_k)\right)^{\gamma^{\beta}(m_k)}}{\prod_{k=1}^{d}\left(1 + v^{\zeta\,q}_{n_{ij}(m_k)}\right)^{\gamma^{\beta}(m_k)} + \prod_{k=1}^{d}\left(1 - v^{\zeta\,q}_{n_{ij}(m_k)}\right)^{\gamma^{\beta}(m_k)}}}\right) \tag{6}$$

**Step 4:**

Define $A^+ = (\alpha^{\aleph+1}, \alpha^{\aleph+2}, \ldots, \alpha^{\aleph+m})^T$ and $A^- = (\alpha^{\aleph-1}, \alpha^{\aleph-2}, \ldots, \alpha^{\aleph-m})^T$ as the "q-ROPF positive ideal solution (q-ROPFPIS) and the q-ROPF negative ideal solution (q-ROPFNIS)" respectively, where $\alpha^{\aleph}+i = (1,0,0), i = 1,2,\ldots,m$ are the m largest q-ROPFNs and $\alpha^{\aleph}-i = (0,1,0), i = 1,2,\ldots,m$ are the m smallest q-ROPFNs. Furthermore, we denote the alternatives $\xi_i^\eta i = 1,2,\ldots,n$ by $\xi_i^\eta = (n_{i1}, n_{i2}, \ldots, r_{im})^T, i = 1,2,\ldots,n$.

**Step 5:**

Calculate the distance between the alternative $\xi_i^\eta$ and the q-ROPFPIS $A^+$ and the distance between the alternative $\xi_i^\eta$ and the q-ROPFNIS $A^-$ respectively:

$$d\left(\xi_i^\eta, A^+\right) = \sum_{j=1}^m \Omega_j d\left(z_{ij}, \xi\eta j+\right)$$

$$= \frac{1}{2} \sum_{j=1}^m \Omega_j \left(\left|\mu^\zeta_{ij} - 1\right| + \left|v^\zeta_{ij} - 0\right| + \left|\pi_{ij} - 0\right|\right)$$

$$= \frac{1}{2} \sum_{j=1}^m \Omega_j \left(1 - \mu\zeta ij + v\zeta ij + \pi ij\right)$$

$$= \frac{1}{2} \sum_{j=1}^m \Omega_j \left(1 - \mu\zeta ij + v\zeta ij + 1 - \mu\zeta ij - v\zeta ij\right)$$

$$= \sum_{j=1}^m \Omega_j \left(1 - \mu\zeta ij\right)$$

$$d\left(\xi_i^\eta, A^-\right) = \sum_{j=1}^m \Omega_j d\left(z_{ij}, \alpha^{\aleph}j-\right)$$

$$= \frac{1}{2} \sum_{j=1}^m \Omega_j \left(\left|\mu^\zeta_{ij} - 0\right| + \left|v^\zeta_{ij} - 1\right| + \left|\pi_{ij} - 0\right|\right)$$

$$= \frac{1}{2} \sum_{j=1}^m \Omega_j \left(1 + \mu^\zeta ij - v^\zeta_{ij} + \pi_{ij}\right)$$

$$= \frac{1}{2} \sum_{j=1}^m \Omega_j \left(1 + \mu^\zeta ij - v^\zeta ij + 1 - \mu^\zeta_{ij} - v^\zeta ij\right)$$

$$= \frac{1}{2} \sum_{j=1}^m \Omega_j \left(1 - v^\zeta ij\right).$$

**Step 6:**

Calculate the closeness coefficient of each alternative:

$$c\left(\xi_i^\eta\right) = \frac{d\left(\xi_i^\eta, A^-\right)}{d\left(\xi_i^\eta, A^+\right) + d\left(\xi_i^\eta, A^-\right)}, \quad i = 1,2,\ldots,n \tag{7}$$

Since

$$d\left(\xi_i^\eta, A^+\right) + d\left(\xi_i^\eta, A^-\right) = \sum_{j=1}^m \Omega_j \left(1 - \mu^\zeta_{ij}\right) + \sum_{j=1}^m \Omega_j \left(1 - v^\zeta_{ij}\right)$$

$$= \sum_{j=1}^m \Omega_j \left(2 - \mu^\zeta_{ij} - v^\zeta_{ij}\right)$$

$$= \sum_{j=1}^m \Omega_j \left(1 + \pi_{ij}\right)$$

Equation (7) can be transformed as:

$$c\left(\xi_i^\eta\right) = \frac{\sum_{j=1}^m \Omega_j \left(1 - v^\zeta_{ij}\right)}{\sum_{j=1}^m \Omega_j \left(1 + \pi_{ij}\right)}, \quad i = 1,2,\ldots,n \tag{8}$$

**Step 7:**

Rank all the alternatives $\xi_i^\eta$ $(i = 1,2,\ldots,n)$ according to the closeness coefficients $c\left(\xi_i^\eta\right)$ $(i = 1,2,\ldots,n)$: the greater the value $c\left(\xi_i^\eta\right)$, the better the alternative $\xi_i^\eta$.

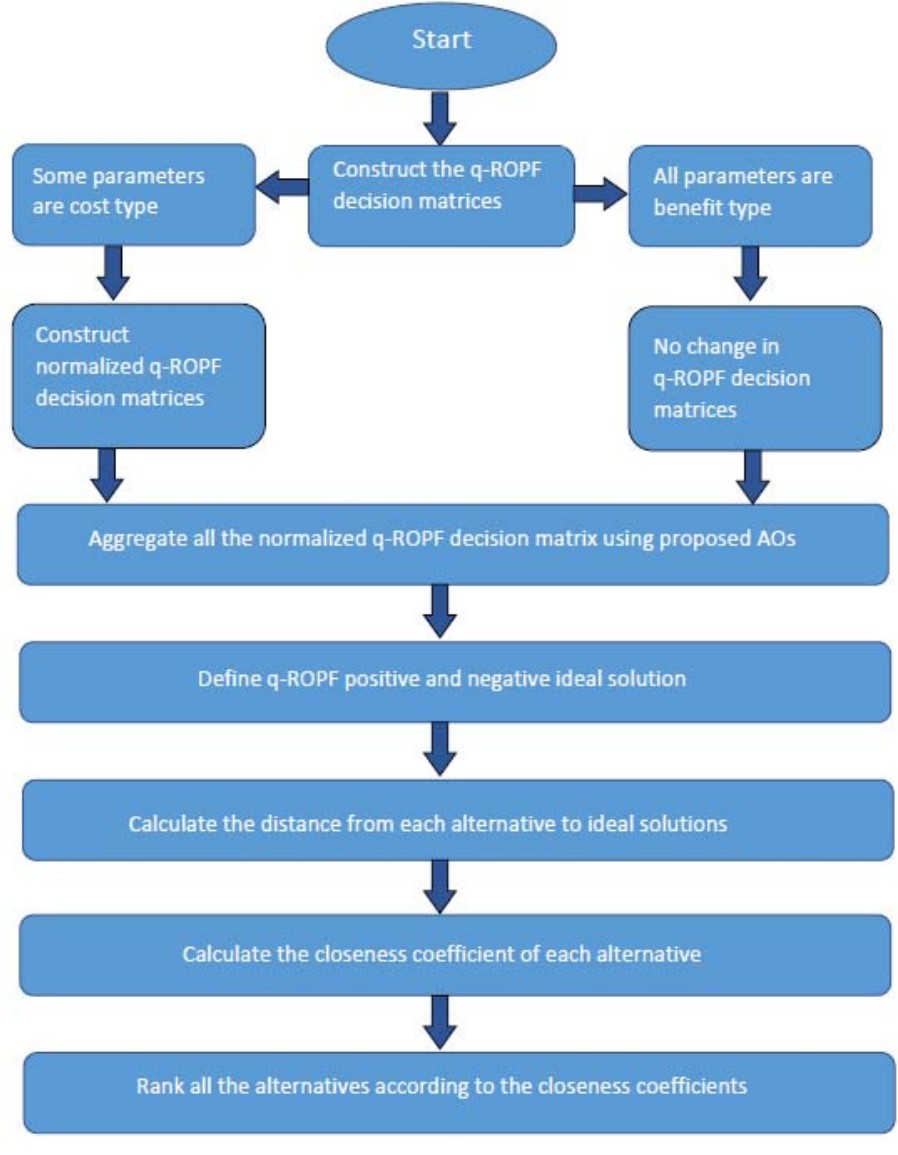

**Figure 2** Pictorial view of proposed Algorithm.

The pictorial view of proposed Algorithm is given in Fig. 2.

## CASE STUDY

The majority of the nation's population lives in metropolitan areas, which are hubs of commercial development and ingenuity. Furthermore, cities' dense population and activity render them susceptible to a variety of pressures, including environmental and man-made calamities. In light of the fact that this is the case, a significant portion of the research that has been carried out over the course of the past several years has focused on the effects that a variety of disasters have had on cities, as well as the essential management, restoration, and adaptive strategies that must be implemented in order to deal with such

disasters. Prior to the outbreak of the COVID-19 epidemic, there was a paucity of study on the relationship between cities and pandemics, despite the fact that this was not the first time in the history of humankind that pandemics had harmed cities. Since the beginning of the COVID-19 emergency, the science-based society has been working tirelessly to shed some light on a variety of issues, including the mechanisms that are driving the spread of the disease, its ecological and cultural consequences, and the retrieval and modification plans and policies that are necessary to address the situation (*Li et al., 2020*). Some of the issues that the science-based society has been working to shed some light on include: Cities are typically designated as "hotspots" for COVID-19 infections because of the high population and economic activity levels that are seen in these places. As a consequence of this, a great number of researchers are striving to study the dynamics of the epidemic in urban places in order to gain a better understanding of the impact that COVID-19 has had on urban populations (*Peng, Zhao & Hu, 2023*).

The features of the surrounding ecosystem can impact the kinetics of dissemination by altering the virus's survival on contaminated sites and/or its airborne spread. The research has examined the effects of many ambient and meteorology characteristics such as weather, dampness, wind velocity, and industrial pollution. Due to the article's urban focus, only outcomes pertaining to the outside environment will be covered (*Xie et al., 2022*). The evidence comes from nations with varying climatic circumstances, including China, Germany, the United States, Argentina, Azerbaijan, Sweden, and Greece. As a consequence of the situationally nature of the study and the large number and sophistication of the elements involved, the results addressing the effect of environmental variables on COVID-19 are not consistent across countries and towns. The pandemic's societal repercussions have been examined in both emerging and advanced countries. While the majority of research focus on negative consequences, there are those that address beneficial social activities sparked by the crisis. The majority of research have concentrated on challenges arising from lengthy structural inequities prevalent in many nations (*Sun et al., 2023b*). Ancient times, epidemics have significantly affected minorities and persons at the bottom of the social ladder. They frequently suffer more from preexisting illnesses as a result of increased risk exposure, financial distress, and restricted access to care. The protracted economic downturn caused by the COVID-19 epidemic has had a devastating effect on the metropolitan economy. The repercussions are diverse and manifest themselves in a variety of ways and at a variety of scales (*Sun et al., 2023a*). While study on this subject is still ongoing, preliminary findings indicate that the epidemic had a substantial impact on city taxation, citizen earnings, entertainment and tourists, medium enterprises, the urban primary sector, and migratory labour. Additionally, a growing body of research has examined the pandemic's consequences' uneven and unequal socioeconomic and regional distribution (*Li, Peng & Wang, 2018*).

## Explanation of problem

Assume a high-level commission has been established to evaluate the influence of COVID-19 on ordinary living in five major cities in a country: $\xi_1^\eta, \xi_2^\eta, \xi_3^\eta, \xi_4^\eta$ and $\xi_5^\eta$. This review panel is composed of members of the COVID-19 board of directors who have been

**Table 2  Assessment matrix acquired for $d_1$.**

|  | $\aleph_1$ | $\aleph_2$ | $\aleph_3$ | $\aleph_4$ | $\aleph_5$ |
|---|---|---|---|---|---|
| $\xi_1^{\eta}$ | (0.657,0.432) | (0.654,0.345) | (0.575,0.155) | (0.435,0.244) | (0.675,0.422) |
| $\xi_2^{\eta}$ | (0.457,0.674) | (0.430,0.340) | (0.355,0.225) | (0.234,0.653) | (0.435,0.765) |
| $\xi_3^{\eta}$ | (0.232,0.321) | (0.234,0.555) | (0.765,0.355) | (0.763,0.256) | (0.665,0.432) |
| $\xi_4^{\eta}$ | (0.754,0.453) | (0.975,0.265) | (0.465,0.543) | (0.245,0.532) | (0.745,0.643) |
| $\xi_5^{\eta}$ | (0.643,0.556) | (0.423,0.465) | (0.265,0.215) | (0.674,0.245) | (0.432,0.653) |

**Table 3  Assessment matrix acquired for $d_2$.**

|  | $\aleph_1$ | $\aleph_2$ | $\aleph_3$ | $\aleph_4$ | $\aleph_5$ |
|---|---|---|---|---|---|
| $\xi_1^{\eta}$ | (0.674,0.245) | (0.535,0.323) | (0.855,0.345) | (0.457,0.355) | (0.425,0.245) |
| $\xi_2^{\eta}$ | (0.355,0.215) | (0.640,0.535) | (0.445,0.570) | (0.745,0.635) | (0.242,0.330) |
| $\xi_3^{\eta}$ | (0.643,0.460) | (0.265,0.335) | (0.235,0.545) | (0.253,0.572) | (0.732,0.225) |
| $\xi_4^{\eta}$ | (0.567,0.754) | (0.255,0.356) | (0.215,0.130) | (0.570,0.562) | (0.879,0.125) |
| $\xi_5^{\eta}$ | (0.341,0.426) | (0.570,0.784) | (0.465,0.532) | (0.674,0.472) | (0.243,0.536) |

**Table 4  Assessment matrix acquired for $d_3$.**

|  | $\aleph_1$ | $\aleph_2$ | $\aleph_3$ | $\aleph_4$ | $\aleph_5$ |
|---|---|---|---|---|---|
| $\xi_1^{\eta}$ | (0.424,0.564) | (0.425,0.265) | (0.575,0.543) | (0.543,0.335) | (0.452,0.543) |
| $\xi_2^{\eta}$ | (0.532,0.356) | (0.345,0.135) | (0.434,0.255) | (0.753,0.320) | (0.424,0.324) |
| $\xi_3^{\eta}$ | (0.424,0.245) | (0.421,0.255) | (0.325,0.890) | (0.753,0.335) | (0.532,0.543) |
| $\xi_4^{\eta}$ | (0.256,0.674) | (0.643,0.365) | (0.455,0.245) | (0.545,0.445) | (0.425,0.254) |
| $\xi_5^{\eta}$ | (0.135,0.356) | (0.575,0.285) | (0.600,0.145) | (0.435,0.245) | (0.573,0.535) |

chosen by the minister of healthcare, commerce, and environment. The panel is tasked with investigating cities depending on five critical criteria. $\aleph_1$ = environmental factors, $\aleph_2$ = urban water cycle, $\aleph_3$ = rate of poverty, $\aleph_4$ =social impacts and $\aleph_5$ =economic impacts throughout the three significant lockdown periods $d_1$, $d_2$ and $d_3$. Assume that $W = (0.10, 0.15, 0.20, 0.25, 0.30)$ represents the weighting of the criterion $\aleph_1$, $\aleph_2$, $\aleph_3$, $\aleph_4$ and $\aleph_5$, and that $\gamma^{\beta}(m_k) = (0.35, 0.30, 0.35)$ represents the weighting of the time periods $d_1$, $d_2$ and $d_3$. Assume experts construct an decision matrix table with dynamic q-ROPFNs. COVID-19 affects the majority of cities in daily life. It is carried out in the following order: Step 1 through Step 7 of Algorithm.

## Decision-making process

**Step 1:** Acquire a decision/assessment matrix $R(m_k) = \left(r_{ij}^k\right)_{m \times n} = \left(\mu^{\zeta'}_{ij}(m_k), v^{\zeta'}_{ij}(m_k)\right)_{m \times n}$ for the $d$ distinct time periods, given in Tables 2, 3 and 4.

Step 2: Normalize the decision matrices acquired by DMs using (5). Here we have two types of criterion. $\aleph_3$ is cost type criteria and others are benefit type criterion. Normalized decision matrices given in Tables 5, 6 and 7.

**Table 5  Normalized assessment matrix for $d_1$.**

|  | $\aleph_1$ | $\aleph_2$ | $\aleph_3$ | $\aleph_4$ | $\aleph_5$ |
|---|---|---|---|---|---|
| $\xi_1^\eta$ | (0.657,0.432) | (0.654,0.345) | (0.155,0.575) | (0.435,0.244) | (0.675,0.422) |
| $\xi_2^\eta$ | (0.457,0.674) | (0.430,0.340) | (0.225,0.355) | (0.234,0.653) | (0.435,0.765) |
| $\xi_3^\eta$ | (0.232,0.321) | (0.234,0.555) | (0.355,0.765) | (0.763,0.256) | (0.665,0.432) |
| $\xi_4^\eta$ | (0.754,0.453) | (0.975,0.265) | (0.543,0.465) | (0.245,0.532) | (0.745,0.643) |
| $\xi_5^\eta$ | (0.643,0.556) | (0.423,0.465) | (0.215,0.265) | (0.674,0.245) | (0.432,0.653) |

**Table 6  Normalized assessment matrix for $d_2$.**

|  | $\aleph_1$ | $\aleph_2$ | $\aleph_3$ | $\aleph_4$ | $\aleph_5$ |
|---|---|---|---|---|---|
| $\xi_1^\eta$ | (0.674,0.245) | (0.535,0.323) | (0.345,0.855) | (0.457,0.355) | (0.425,0.245) |
| $\xi_2^\eta$ | (0.355,0.215) | (0.640,0.535) | (0.570,0.445) | (0.745,0.635) | (0.242,0.330) |
| $\xi_3^\eta$ | (0.643,0.460) | (0.265,0.335) | (0.545,0.235) | (0.253,0.572) | (0.732,0.225) |
| $\xi_4^\eta$ | (0.567,0.754) | (0.255,0.356) | (0.130,0.215) | (0.570,0.562) | (0.879,0.125) |
| $\xi_5^\eta$ | (0.341,0.426) | (0.570,0.784) | (0.532,0.465) | (0.674,0.472) | (0.243,0.536) |

**Table 7  Normalized assessment matrix for $d_3$.**

|  | $\aleph_1$ | $\aleph_2$ | $\aleph_3$ | $\aleph_4$ | $\aleph_5$ |
|---|---|---|---|---|---|
| $\xi_1^\eta$ | (0.424,0.564) | (0.425,0.265) | (0.543,0.575) | (0.543,0.335) | (0.452,0.543) |
| $\xi_2^\eta$ | (0.532,0.356) | (0.345,0.135) | (0.255,0.434) | (0.753,0.320) | (0.424,0.324) |
| $\xi_3^\eta$ | (0.424,0.245) | (0.421,0.255) | (0.890,0.325) | (0.753,0.335) | (0.532,0.543) |
| $\xi_4^\eta$ | (0.256,0.674) | (0.643,0.365) | (0.245,0.455) | (0.545,0.445) | (0.425,0.254) |
| $\xi_5^\eta$ | (0.135,0.356) | (0.575,0.285) | (0.145,0.600) | (0.435,0.245) | (0.573,0.535) |

**Step 3:**

In this step we utilized proposed DQROPFEWA operator to aggregate all the normalized decision matrices $N(m_k) = \left(n_{ij}^k\right)_{m \times n} = \left(\mu_{ij}^\zeta(m_k), \nu_{ij}^\zeta(m_k)\right)_{m \times n}$ into one cumulative q-ROPF matrix $Z = (z_{ij})_{m \times n} = \left(\mu_{ij}^\zeta, \nu_{ij}^\zeta\right)_{m \times n}$, given in Table 8.

**Step 4:**

Define $A^+ = (\alpha^{\aleph+1}, \alpha^{\aleph+2}, \ldots, \alpha^{\aleph+m})^T$ and $A^- = (\alpha^{\aleph-1}, \alpha^{\aleph-2}, \ldots, \alpha^{\aleph-m})^T$ as the q-ROPF positive ideal solution (q-ROPFPIS) and the q-ROPF negative ideal solution (q-ROPFNIS) as

$A^+ = \Big((1,0,0),(1,0,0),(1,0,0),(1,0,0),(1,0,0)\Big),$

$A^- = \Big((0,1,0),(0,1,0),(0,1,0),(0,1,0),(0,1,0)\Big).$

and

$\xi_1^\eta = \Big((0.603705, 0.402249, 0.894200),$
$(0.556216, 0.308500, 0.927800), (0.412201, 0.654471, 0.866100),$
$(0.484348, 0.305253, 0.950200), (0.549099, 0.393372, 0.918000)\Big)$

**Table 8  Aggregated values.**

|  | $\aleph_1$ | $\aleph_2$ | $\aleph_3$ | $\aleph_4$ | $\aleph_5$ |
|---|---|---|---|---|---|
| $\xi_1^{\eta}$ | (0.603705, 0.402249) | (0.556216, 0.308500) | (0.412201, 0.654471) | (0.484348, 0.305253) | (0.549099, 0.393372) |
| $\xi_2^{\eta}$ | (0.463707, 0.387514) | (0.496206, 0.283795) | (0.403923, 0.328960) | (0.660179, 0.509706) | (0.391021, 0.448084) |
| $\xi_3^{\eta}$ | (0.482519, 0.325964) | (0.330856, 0.365182) | (0.708360, 0.363422) | (0.683976, 0.359841) | (0.651186, 0.386867) |
| $\xi_4^{\eta}$ | (0.601855, 0.611763) | (0.837205, 0.324042) | (0.396317, 0.367519) | (0.490435, 0.508572) | (0.739610, 0.288521) |
| $\xi_5^{\eta}$ | (0.475750, 0.440266) | (0.530132, 0.466380) | (0.368453, 0.420480) | (0.612943, 0.299050) | (0.462450, 0.574853) |

$$\xi_2^{\eta} = \Big((0.463707, 0.387514, 0.944300),$$
$$(0.496206, 0.283795, 0.949100), (0.403923, 0.328960, 0.965000),$$
$$(0.660179, 0.509706, 0.833900), (0.391021, 0.448084, 0.947400)\Big)$$

$$\xi_3^{\eta} = \Big((0.482519, 0.325964, 0.948400), (0.330856, 0.365182, 0.970900),$$
$$(0.708360, 0.363422, 0.841800), (0.683976, 0.359841, 0.858800),$$
$$(0.651186, 0.386867, 0.873300)\Big)$$

$$\xi_4^{\eta} = \Big((0.601855, 0.611763, 0.820800), (0.837205, 0.324042, 0.723800),$$
$$(0.396317, 0.367519, 0.965900), (0.490435, 0.508572, 0.908800),$$
$$(0.739610, 0.288521, 0.829800)\Big)$$

$$\xi_5^{\eta} = \Big((0.475750, 0.440266, 0.931000), (0.530132, 0.466380, 0.908400),$$
$$(0.368453, 0.420480, 0.956700), (0.612943, 0.299050, 0.905700),$$
$$(0.462450, 0.574853, 0.892600)\Big)$$

**Step 5 and Step 6:**

Calculate the distance between the alternative $\xi_i^{\eta}$ and the q-ROPFPIS $A^+$ and the distance between the alternative $\xi_i^{\eta}$ and the q-ROPFNIS $A^-$ respectively. Then we calculate the closeness coefficient of each alternative:

$$c(\xi_1^{\eta}) = 0.307235$$
$$c(\xi_2^{\eta}) = 0.307432$$
$$c(\xi_3^{\eta}) = 0.336204$$
$$c(\xi_4^{\eta}) = 0.324206$$
$$c(\xi_5^{\eta}) = 0.289676$$

**Step 7:**

Rank all the alternatives $\xi_i^{\eta} i = 1, 2, \ldots, n$ according to the closeness coefficients $c\left(\xi_i^{\eta}\right)$ $(i = 1, 2, \ldots, n)$

$$\xi_3^{\eta} > \xi_4^{\eta} > \xi_2^{\eta} > \xi_1^{\eta} > \xi_5^{\eta}.$$

## Limitations of the proposed method

In order to highlight the inadequacy of the presented methodologies, we undertake a rigorous study of the Algorithm and identify its flaws.

- In the examples that came before, logical relationships between the arguments were not taken into account.

**Table 9  Comparison of proposed operators with some exiting operators.**

| Authors | AOs | Ranking of alternatives | The optimal alternative |
|---|---|---|---|
| *Riaz et al. (2020)* | q-ROPFEWA | $\xi_3^\eta > \xi_4^\eta > \xi_1^\eta > \xi_2^\eta > \xi_5^\eta$ | $\xi_3^\eta$ |
| | q-ROPFEOWA | $\xi_3^\eta > \xi_4^\eta > \xi_2^\eta > \xi_5^\eta > \xi_1^\eta$ | $\xi_3^\eta$ |
| *Liu & Wang (2018)* | q-ROPFWA | $\xi_3^\eta > \xi_4^\eta > \xi_2^\eta > \xi_1^\eta > \xi_5^\eta$ | $\xi_3^\eta$ |
| | q-ROPFWG | $\xi_3^\eta > \xi_4^\eta > \xi_2^\eta > \xi_5^\eta > \xi_1^\eta$ | $\xi_3^\eta$ |
| *Jana, Muhiuddin & Pal (2019)* | q-ROPFDWA | $\xi_3^\eta > \xi_4^\eta > \xi_2^\eta > \xi_1^\eta > \xi_5^\eta$ | $\xi_3^\eta$ |
| | q-ROPFDWG | $\xi_3^\eta > \xi_4^\eta > \xi_2^\eta > \xi_5^\eta > \xi_1^\eta$ | $\xi_3^\eta$ |
| *Peng, Dai & Garg (2018)* | q-ROPFEWA | $\xi_3^\eta > \xi_4^\eta > \xi_2^\eta > \xi_1^\eta > \xi_5^\eta$ | $\xi_3^\eta$ |
| | q-ROPFEWG | $\xi_3^\eta > \xi_1^\eta > \xi_2^\eta > \xi_4^\eta > \xi_5^\eta$ | $\xi_3^\eta$ |
| *Farid & Riaz (2023)* | q-ROPFAAWA | $\xi_3^\eta > \xi_4^\eta > \xi_2^\eta > \xi_1^\eta > \xi_5^\eta$ | $\xi_3^\eta$ |
| | q-ROPFAAWG | $\xi_3^\eta > \xi_1^\eta > \xi_2^\eta > \xi_4^\eta > \xi_5^\eta$ | $\xi_3^\eta$ |
| Proposed | DQROPFEWA | $\xi_3^\eta > \xi_4^\eta > \xi_2^\eta > \xi_1^\eta > \xi_5^\eta$ | $\xi_3^\eta$ |
| | DQROPFEWG | $\xi_3^\eta > \xi_2^\eta > \xi_4^\eta > \xi_5^\eta > \xi_1^\eta$ | $\xi_3^\eta$ |

- It is not typically accurate to assert that each parameter in the MPDM may be considered independent of the others when working with real data. Any one of the MPDM's parameters might be dependent on or related to a different set of parameters.
- The objectivity of judgements made using the offered MPDM approaches should be improved by the evaluation of the interdependence between parameters. It's possible that the quality of the decision-making framework might be improved by taking reliance into account in the q-ROPF MPDM.

## Comparative study

In this part of the article, we will contrast the proposed operators with specific AOs that are already being utilised. The fact that both of these approaches end up with the same outcome indicates why our proposed AOs are preferable. By resolving the information data with several AOs that are already in use, we are able to compare our results and arrive at the same optimal conclusion. This illustrates both the soundness and coherence of the paradigm that we proposed. We receive a rating of $\xi_3^\eta > \xi_4^\eta > \xi_2^\eta > \xi_1^\eta > \xi_5^\eta$ from our suggested AOs; in order to confirm our best choice, we analyse this problem using other AOs that are already in place. The fact that we both arrive at the same option that is optimal suggests that the AOs that we have provided are correct. The Table 9 compares the AOs available with certain existing AOs.

## CONCLUSION

To aggregate the q-ROPF information acquired over time, several dynamic q-rung orthopair fuzzy AOs are developed. These include the dynamic q-rung orthopair fuzzy Einstein weighted averaging (DQROPFEWA) operator and the dynamic q-rung orthopair fuzzy Einstein weighted geometric (DQROPFEWG) operator. All of the operators include time into the aggregate process, and hence are time dependent operators, which solve some of the shortcomings of conventional static q-ROPF aggregation operators. The suggested operators are shown to have a number of desired features. Additionally, using the suggested

dynamic q-ROPF operators, we proposed a method for solving MPDM issues in which all decision information is in the form of q-ROPFNs acquired over time. Finally, an example is shown to demonstrate the suggested dynamic operators and established technique. Later, a comparative analysis with the previous research results was conducted to determine the efficiency of the suggested approach. The primary advantage of this suggested strategy is that it is more broad than others in terms of accumulating q-ROPF information. The proposed method can be used to develop future dynamic decision-making methods such as light field depth estimation (*Cui et al., 2023*), multiscale feature extraction (*Lu et al., 2023*), adapting feature selection (*Liu et al., 2023*), situation-aware dynamic service (*Cheng et al., 2017*) and random optimization (*Zhang et al., 2023*).

### Funding
The authors received no funding for this work.

### Competing Interests
Vladimir Simic is an Academic Editor for PeerJ.

### Author Contributions
- Hafiz Muhammad Athar Farid conceived and designed the experiments, analyzed the data, prepared figures and/or tables, and approved the final draft.
- Muhammad Riaz performed the experiments, prepared figures and/or tables, and approved the final draft.
- Vladimir Simic conceived and designed the experiments, analyzed the data, performed the computation work, authored or reviewed drafts of the article, and approved the final draft.
- Xindong Peng performed the experiments, authored or reviewed drafts of the article, and approved the final draft.

### Data Availability
The raw data is available in the tables.

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
