# Peer review of "q-Rung orthopair fuzzy dynamic aggregation operators with time sequence preference for dynamic decision-making"

_PeerJ Computer Science, doi:10.7717/peerj-cs.1742_

## Round 0.1 · original submission · Major Revisions

The paper needs to be improved in several respects, among which:

1) the presentation needs improvement from a linguistic point of view

2) both the motivations and the novelty of the proposed approach have to be better outlined

3) some formal aspects have to be clarified, as outlined by the reviewers.

**Language Note:** The Academic Editor has identified that the English language must be improved. PeerJ can provide language editing services - please contact us at copyediting@peerj.com for pricing (be sure to provide your manuscript number and title). Alternatively, you should make your own arrangements to improve the language quality and provide details in your response letter. – PeerJ Staff

Reviewer 1 ·

Basic reporting

N/a

Experimental design

N/a

Validity of the findings

N/a

Additional comments

N/a

Reviewer 2 ·

Basic reporting

The goal of this study is to introduce some dynamic q-rung orthopair fuzzy aggregation operators (AOs) for solving multi-period decision-making issues in which all decision information is given by decision makers in the form of q-rung orthopair fuzzy numbers" (q-ROFNs) spanning diverse time periods. Einstein AOs are used to provide seamless information fusion, taking this advantage the authors proposed two new AOs namely, dynamic q-rung orthopair fuzzy Einstein weighted averaging (DQROFEWA) operator and dynamic q-rung orthopair fuzzy Einstein weighted geometric (DQROFEWG) operator". Several attractive features of these AOs are addressed in depth. Additionally, we develop a method for addressing multi-period decision-making problems by using ideal solutions. To demonstrate the suggested approach's use, a numerical example is provided for calculating the impact of coronavirus disease" 2019 (COVID-19) on everyday living. Finally, a comparison of the proposed and existing studies is performed to establish the efficacy of the proposed method.

Experimental design

See above.

Validity of the findings

See above.

Reviewer 3 ·

Basic reporting

In this paper, authors introduced some dynamic q-rung orthopair fuzzy aggregation operators for solving multi-period decision-making issues in which all decision information is given by decision makers in the form of q-rung orthopair fuzzy numbers spanning diverse time periods. Paper is written well and can be published after some improvements, my concerns are as follows.
1. The motivation of this study is not clearly illustrated. You should make a clear discussion about the literature and show the challenges that need to be tackled.
2. The linguistic quality should be improved to guarantee a smooth reading.
3. The author should analyze the relationship between DQROFEWA operator and DQROFEWG operator.
4. It is obvious that the weights of attributes and experts will influence the decision-making results. Authors should analyze the decision-making methods if the weights of attributes and experts are complete unknown or partially known in detail.
5. The authors had better enhance the discussion for limitations of this proposed approaches and the future directions in the conclusion section.
6. The data of in Section 5.2, how did you get these data? Or what is the basis to set values of these indicators.
7. Put the proof of Theorem 3.5 and Theorem 3.6.
8. Please enhance the compassion between the proposed method with other methods.

Experimental design

The data of in Section 5.2, how did you get these data? Or what is the basis to set values of these indicators.

Validity of the findings
* * *
Additional comments
* * *
Reviewer 4 ·

Basic reporting

Authors developed some dynamic aggregation operators for the data spanning diverse time periods. Several attractive features of these AOs are addressed in depth. Additionally, they developed method for addressing multi-period decision-making problems by using ideal solutions. Paper can be published after some corrections.
1. Rewrite the abstract from the beginnings by focus on the "idea, develop points in this method presented by author"s" and compare the results of this manuscript with other papers as values.
2. What's the differentiation between proposed method and previous method? It is not clear in introduction.
3. Multi points in this manuscript need scientific justification and prove by the author(s), after each table or figure give specific description of result submitted by it described in three into five lines.
4. Some references are not included in the study. The authors should include references to the places where question marks are given in the study. Can author compare results with 1. "New group-based generalized interval-valued q-rung orthopair fuzzy soft aggregation operators and their applications in sports decision-making problems", 2. "Aggregation and interaction aggregation soft operators on interval-valued q-rung orthopair fuzzy soft environment and application in automation company evaluation", 3. Group Generalized q-Rung Orthopair Fuzzy Soft Sets: New Aggregation Operators and Their Applications
5. Many dynamic aggregation operators in the literature have been developed by various researchers. What are the differences between this study and other studies? Several dynamic MCDM studies deal with dynamic aggregation operators. Why were Einstein operators used in the study?
6. The font of Figure 1 does not match the manuscript font. The font of figure 1 should be corrected.
7. Why were q-ROFPIS and q-ROFNIS used with 1s and 0s in the study? Why was it not considered by considering the smallest and largest values.

Experimental design

Comments are given above

Validity of the findings

Comments are given above

---

## Round 0.2 · accepted · Accept

Dear authors,

The revised version of your paper has addressed the recommendations and the issues raised by the first-round reviewers. It can then be accepted for publication on PeerJ Computer Science.

Reviewer 3 ·

Basic reporting

I want to express my gratitude to the authors for addressing my concerns. I am pleased to inform you that, in its current form, I believe the manuscript is suitable for acceptance.

Experimental design

--

Validity of the findings

Confirmed

Additional comments

--

Reviewer 4 ·

Basic reporting

The paper can be accepted in current form.

Experimental design

fine.

Validity of the findings

fine

Additional comments

The paper can be accepted in current form.